# Not All Patches are What You Need: Expediting Vision Transformers via Token Reorganizations

**Youwei Liang**[1][*]  **Chongjian Ge**[2]  **Zhan Tong**[3]  **Yibing Song**[3]  **Jue Wang**[3]  **Pengtao Xie**[1][†]

[1]UC San Diego    [2]The University of Hong Kong    [3]Tencent AI Lab
youwei@ucsd.edu    rhettgee@connect.hku.hk    elliottong@tencent.com
yibingsong.cv@gmail.com    arphid@gmail.com    p1xie@eng.ucsd.edu

## ABSTRACT

Vision Transformers (ViTs) take all the image patches as tokens and construct multi-head self-attention (MHSA) among them. Complete leverage of these image tokens brings redundant computations since not all the tokens are attentive in MHSA. Examples include that tokens containing semantically meaningless or distractive image backgrounds do not positively contribute to the ViT predictions. In this work, we propose to reorganize image tokens during the feed-forward process of ViT models, which is integrated into ViT during training. For each forward inference, we identify the attentive image tokens between MHSA and FFN (i.e., feed-forward network) modules, which is guided by the corresponding class token attention. Then, we reorganize image tokens by preserving attentive image tokens and fusing inattentive ones to expedite subsequent MHSA and FFN computations. To this end, our method EViT improves ViTs from two perspectives. First, under the same amount of input image tokens, our method reduces MHSA and FFN computation for efficient inference. For instance, the inference speed of DeiT-S is increased by 50% while its recognition accuracy is decreased by only 0.3% for ImageNet classification. Second, by maintaining the same computational cost, our method empowers ViTs to take more image tokens as input for recognition accuracy improvement, where the image tokens are from higher resolution images. An example is that we improve the recognition accuracy of DeiT-S by 1% for ImageNet classification at the same computational cost of a vanilla DeiT-S. Meanwhile, our method does not introduce more parameters to ViTs. Experiments on the standard benchmarks show the effectiveness of our method. The code is available at https://github.com/youweiliang/evit

## 1 INTRODUCTION

Computer vision research has evolved into Transformers since ViTs (Dosovitskiy et al., 2021). Equipped with global self-attention, ViTs have shown impressive capability upon local convolution (i.e., CNNs) on prevalent visual recognition scenarios, including image classification (Dosovitskiy et al., 2021; Touvron et al., 2021a; Jiang et al., 2021; Graham et al., 2021), object detection (Carion et al., 2020), and semantic segmentation (Xie et al., 2021; Liu et al., 2021; Wang et al., 2021a;c), with both supervised and unsupervised (self-supervised) training (Pan et al., 2021; Ge et al., 2021) configurations. Based on the main spirit of ViTs (i.e., MHSA), there are wide investigations (Liu et al., 2021; Chu et al., 2021; Wang et al., 2021a) to explore the network structure of ViT models for continuous recognition performance improvement.

Along with the development of ViT models, the computation burden is becoming an issue. As illustrated in (Dosovitskiy et al., 2021), training a ViT from scratch typically requires larger datasets (e.g., ImageNet-21k (Deng et al., 2009) and JFT-300M (Sun et al., 2017)) than those of CNNs (e.g.,

---

[*]Major work done during an internship at Tencent AI Lab.
[†]Corresponding author.

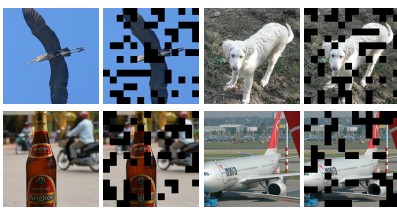
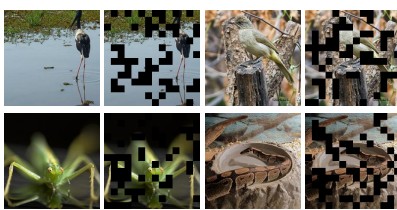

|                        |                     |
| :--------------------: | :-----------------: |
| (a) Mask on backgrounds | (b) Mask on objects |

Figure 1: DeiT-S (Touvron et al., 2021a) predictions with incomplete input image tokens. In (a), removing image tokens unrelated to the visual content of the corresponding category does not deteriorate ViT predictions. In (b), removing related image tokens makes ViT predict incorrectly.

CIFAR-10/100 (Krizhevsky et al., 2009) and ImageNet-1k). Also, using more training iterations is a necessity for network convergence (Dosovitskiy et al., 2021; Touvron et al., 2021a; Wang et al., 2022). Without such large scale training, the ViT models are not fully exploited and perform inferior on visual recognition scenarios. These issues motivate us to expedite ViTs for practice usage.

The model acceleration of ViTs is important to reduce computational complexity. However, there are few studies focused on ViT acceleration. This is because the significant model difference between CNNs and ViTs prevents CNN model acceleration (e.g., pruning and distillation) from applying on ViTs. Nevertheless, we analyze ViT from another perspective. We observe that not all image tokens in ViTs contribute positively to the final predictions. Fig. 1 shows some examples where part of the input image tokens are randomly dropped out. In Fig. 1b, removing image tokens related to the visual content of the corresponding category makes the ViT predict incorrectly. In comparison, removing unrelated image tokens does not affect ViT predictions, as shown in Fig. 1a. On the other hand, we notice that ViTs divide images into non-overlapping tokens and perform self-attention (Vaswani et al., 2017) computation on these tokens. A notable characteristic of self-attention is that it can process a varying number of tokens. These observations motivate us to reorganize image tokens for ViT model accelerations. Naseer et al. (2021) also showed that ViT are robust to patch drop, which motivates the idea of dropping the less informative patches to accelerate ViT inference.

In this work, we propose a token reorganization method to identify and fuse image tokens. Given all the image tokens as input, we compute token attentiveness between these tokens and the class token for identification. Then, we preserve the attentive image tokens and fuse the inattentive tokens into one token to allow the gradient back-propagate through the inattentive tokens for better attentive token identification. In this way, we gradually reduce the number of image tokens as the network goes deeper to decrease computation cost. Also, the capacity of the ViT backbone can be flexibly controlled via the identification process where no additional parameters are introduced. We adopt our token reorganization method on representative ViT models (i.e., DeiT (Touvron et al., 2021a) and LV-ViT (Jiang et al., 2021)) for ImageNet classification evaluation. The experimental results show our advantages. For instance, with the same amount of input image tokens, our method speeds up the DeiT-S model by 50%, while only sacrificing 0.3% recognition accuracy on ImageNet classification. On the other hand, we extend our methods to boost the ViT model recognition performance under the same computational cost. By increasing the input image resolution, our method facilitates Vision Transformers in taking more image tokens to achieve higher classification accuracy. Numerically, we improve the ImageNet classification accuracy of the DeiT-S model by 1% under the same computational cost. Moreover, by using an oracle ViT to guide the token reorganization process, our method can increase the accuracy of the original DeiT-S from 79.8% to 80.7% while reducing its computation cost by 36% under the multiply-accumulate computation (MAC) metric.

## 2 RELATED WORK

### 2.1 VISION TRANSFORMERS

Transformers (Vaswani et al., 2017) have drawn much attention to computer vision recently due to its strong capability of modeling long-range relation. A few attempts have been made to add self-attention layers or Transformers on top of CNNs in image classification (Hu et al., 2019), object de-

tection (Carion et al., 2020), segmentation (Wang et al., 2021c), image retrieval (Lu et al., 2019) and even video understanding (Sun et al., 2019; Girdhar et al., 2019). Vision Transformer (ViT) (Dosovitskiy et al., 2021) first introduced a set of pure Transformer backbones for image classification and its follow-ups modify the ViT architecture for not only better visual recognition (Touvron et al., 2021a; Yuan et al., 2021; Zhou et al., 2021) but many other high-level vision tasks, such as object detection (Carion et al., 2020; Zhu et al., 2020; Liu et al., 2021), semantic segmentation (Wang et al., 2021a; Xie et al., 2021; Chu et al., 2021), and video understanding (Bertasius et al., 2021; Fan et al., 2021). Vision Transformers have shown its strong potential as an alternative to the previously dominant CNNs.

## 2.2 MODEL ACCELERATION

Neural networks are typically overparameterized (Allen-Zhu et al., 2019), which results in significant redundancy in computation in deep learning models. To deploy deep neural networks on mobile devices, we must reduce the storage and computational overhead of the networks. Many adaptive computation methods are explored (Bengio et al., 2015; Wang et al., 2018; Graves, 2016; Hu et al., 2020; Wang et al., 2020b; Han et al., 2021b) to alleviate the computation burden. Parameter pruning (Srinivas & Babu, 2015; Han et al., 2015; Chen et al., 2015b) reduces redundant parameters which are not sensitive to the final performance. Some other methods leverage knowledge distillation (Hinton et al., 2015; Romero et al., 2014; Luo et al., 2016; Chen et al., 2015a) to obtain a small and compact model with distilled knowledge of a larger one. These model acceleration strategies are limited to convolutional neural networks.

There are also some attempts to accelerate the computation of the Transformer model, including proposing more efficient attention mechanisms (Wang et al., 2020a; Kitaev et al., 2020; Choromanski et al., 2020) and the compressed Transformer structures (Liu et al., 2021; Heo et al., 2021; Wang et al., 2021a). These methods mainly focus on reducing the complexity of the network architecture through artificially designed modules. Another approach to ViT acceleration is reducing the number of tokens involved in the inference of ViTs. Notably, Wang et al. (2021b) proposed a method to dynamically determine the number of patches to divide on an image. The ViT will stop inference for an input image if it has sufficient confidence in the prediction of the intermediate outputs. Remarkably, Ryoo et al. (2021) proposed TokenLearner to expedite ViTs, where a relatively small amount of tokens are learned by aggregating the entire feature map weighted by a dynamic attention map conditioned on the feature map. This can be seen as a sophisticated method for tokenizing the input images. Different from TokenLearner, our work focuses on the progressive selection of informative tokens during training. Another related work is DynamicViT (Rao et al., 2021), which introduces a method to reduce token for a fully trained ViT, where an extra learnable neural network is added to ViT to select a subset of tokens. Our work provides a novel perspective for reducing the computational overhead of inference by proposing a token reorganization method to progressively reduce and reorganize image tokens. Unlike DynamicViT, our method does not need a fully trained ViT to help the training and brings no additional parameters into ViT.

## 3 TOKEN REORGANIZATIONS

Our method EViT is built upon ViT (Dosovitskiy et al., 2021) and its variants for visual recognition. We first review ViT and then present how to incorporate our method into the ViT training procedure. Each component of EViT, including attentive token identification and inattentive token fusion, will be elaborated. Furthermore, we analyze the effectiveness of our method by visualizing the attentive tokens at different layers and discuss training on higher resolution images with EViT.

### 3.1 VIT OVERVIEW

Vision Transformers (ViTs) are first introduced by Dosovitskiy et al. (2021) into visual recognition. They perform tokenization by dividing an input image into patches and projecting each patch to a token embedding. An extra class token `[CLS]` is added to the set of image tokens and is responsible for aggregating global image information and final classification. All of the tokens are added by a learnable vector (i.e., positional encoding) and fed into the sequentially-stacked Transformer encoders consisting of a multi-head self-attention (MHSA) layer and a feed-forward network (FFN).

| Token keep rate | 1.0 | 0.9 | 0.8 | 0.7 | 0.6 | 0.5 |
|---|---|---|---|---|---|---|
| Top-1 Acc (%) | 79.8 | 79.7(-0.1) | 79.2(-0.6) | 78.5(-1.3) | 76.8(-3.0) | 73.8(-6.0) |

Table 1: ImageNet classification accuracy of a straightforward inattentive token removal for a trained DeiT-S (Touvron et al., 2021a). The inattentive tokens are directly removed based on the attention from the class token to other tokens at the $4^{th}$, $7^{th}$ and $10^{th}$ layers.

In MHSA, the tokens are linearly mapped and further packed into three matrices, namely $\boldsymbol{Q}, \boldsymbol{K}$, and $\boldsymbol{V}$. The attention operation is conducted as follows.

$$\text{Attention}(\boldsymbol{Q}, \boldsymbol{K}, \boldsymbol{V}) = \text{Softmax}(\frac{\boldsymbol{Q}\boldsymbol{K}^\top}{\sqrt{d}})\boldsymbol{V}. \tag{1}$$

where $d$ is the length of the query vector. The result of $\text{Softmax}(\boldsymbol{Q}\boldsymbol{K}^\top/\sqrt{d})$ is a square matrix which is called the attention map. The first row of attention map represents the attention from [CLS] to all tokens and will be used to determine the attentiveness (importance) of each token (detailed in the next subsection). The output tokens of MHSA are sent to FFN, consisting of two fully connected layers with a GELU activation layer (Hendrycks & Gimpel, 2016) in between. At the final Transformer encoder layer, the [CLS] token is extracted and utilized for object category prediction. More details of Transformers can be found in Vaswani et al. (2017).

## 3.2 ATTENTIVE TOKEN IDENTIFICATION

Let $n$ denote the number of input tokens to a ViT encoder. In the last encoder of ViT, the [CLS] token is taken out for classification. The interactions between [CLS] and other tokens are performed via the attention mechanism (Vaswani et al., 2017) in the ViT encoders:

$$\boldsymbol{x}_{\text{class}} = \text{Softmax}(\frac{\boldsymbol{q}_{\text{class}} \cdot \boldsymbol{K}^\top}{\sqrt{d}})\boldsymbol{V} = \boldsymbol{a} \cdot \boldsymbol{V}. \tag{2}$$

where $\boldsymbol{q}_{\text{class}}$, $\boldsymbol{K}$, and $\boldsymbol{V}$ denote the query vector of [CLS], the key matrix, and the value matrix, respectively, in an attention head. In other words, the output of the [CLS] token $\boldsymbol{x}_{\text{class}}$ is a linear combination of the value vectors $\boldsymbol{V} = [\boldsymbol{v}_1, \boldsymbol{v}_2, \ldots, \boldsymbol{v}_n]^\top$, with the combination coefficients (denoted by $\boldsymbol{a}$ in Eq. 2) being the attention values from [CLS] with respect to all tokens. Since $\boldsymbol{v}_i$ comes from the $i$-th token, the attention value $a_i$ (i.e., the $i$-th entry in $\boldsymbol{a}$) determines how much information of the $i$-th token is fused into the output of [CLS] (i.e., $\boldsymbol{x}_{\text{class}}$) through the linear combination. It is thus natural to assume that the attention value $a_i$ indicates the importance of the $i$-th token.

Moreover, Caron et al. (2021) also showed that the [CLS] token in ViTs pays more attention (i.e., having a larger attention value) to class-specific tokens than to the tokens on the non-object regions. To this end, we propose to use the attentiveness of the [CLS] token with respect to other tokens to identify the most important tokens. Based on these arguments, a simple method to reduce computation in ViT is to remove the tokens with the smallest attention values. However, we find that directly removing those tokens severely deteriorates the classification accuracy, as shown in Table 1. Therefore, we propose to incorporate image token reorganization during the ViT training process.

In multi-head self-attention layer, there are multiple heads performing the computation of Eq. 1 in parallel. Thus, there are multiple [CLS] attention vectors $\boldsymbol{a}^{(h)}$, $h = [1, \ldots, H]$, with $H$ being the total number of attention heads (Vaswani et al., 2017). We compute the average attentiveness value of all heads by $\bar{\boldsymbol{a}} = \sum_{h=1}^{H} \boldsymbol{a}^{(h)}/H$. As shown in Figure 2, we identify and preserve the tokens corresponding to the $k$ largest (top-k) elements in $\bar{\boldsymbol{a}}$ ($k$ is a hyperparameter), which we call the attentive tokens, and further fuse the other tokens (which we call the inattentive tokens) into a new token. The fusion of tokens is detailed in the following paragraph. We define the token keeping rate as $\kappa = k/n$.

## 3.3 INATTENTIVE TOKEN FUSION

Although the tokens on the backgrounds of images are less informative and can be discarded without significantly influencing the performance of the ViT model, they may still be able to contribute to

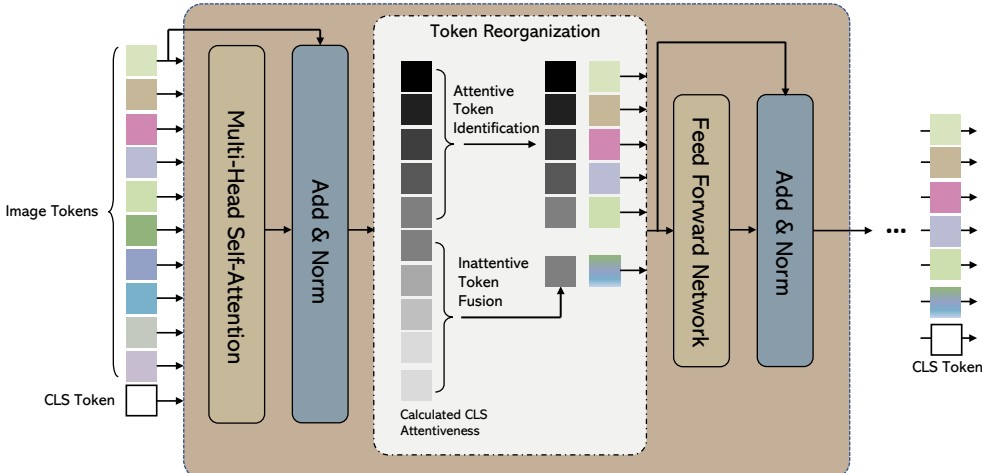

Figure 2: Token reorganization within a single Transformer encoder. Based on ViT (Dosovitskiy et al., 2021), we reorganize tokens in the original Transformer encoder. Specifically, we calculate the attentiveness of the class token with respect to each image token. Then, we use the attentiveness value as a criterion to identify the top-k attentive tokens and fuse the inattentive tokens.

the prediction results. On the other hand, some images may have large object parts covering a large proportion of the images. Thus, it is possible to remove some tokens corresponding to those object parts when we select a fixed number of tokens to keep in a ViT encoder, which may negatively affect the image recognition performance. To mitigate these problems, we propose to fuse the inattentive tokens at the current stage to supplement attentive ones, as illustrated in Figure 2. The inattentive token fusion benefits our method to preserve more information provided by the inattentive tokens. Specifically, we denote the indices set of the inattentive tokens as $\mathcal{N}$. The proposed inattentive token fusion is a weighted average operation, which can be written as $\boldsymbol{x}_{\text{fused}} = \sum_{i \in \mathcal{N}} a_i \boldsymbol{x}_i$. The fused token $\boldsymbol{x}_{\text{fused}}$ is appended to the attentive tokens and sent to the subsequent layers. The computation cost of token fusion is negligible compared to the bulk computation of ViT.

## 3.4 ANALYSIS

**Training with higher resolution images.** Since our approach is efficient in processing image tokens, we are able to feed more tokens into an EViT while maintaining the computational cost at the same level as a vanilla ViT. A straightforward method to get more tokens is resizing the input images to a higher resolution and keeping the patch size unchanged. Note that these higher resolution images do not necessarily come from raw images with a higher resolution. In our vision recognition experiments on ImageNet, we simply resize the standard input images of size $224 \times 224$ to a larger spatial size (e.g., $256 \times 256$) via bicubic interpolation to obtain the higher resolution images, which are further divided into more tokens. Therefore, compared to a vanilla ViT, EViT uses *no* additional information from the images to obtain the prediction results in both training and inference. The experimental results in Table 5 validate the effectiveness of the proposed method.

**Visualization.** The proposed EViT accelerates ViTs by identifying the attentive tokens and discarding the redundant calculation on inattentive image tokens. To further investigate the interpretability of EViT, we visualize the attentive token identification procedure in Figure 3. We present the original images and the attentive token identification results at different layers (e.g., the $4^{th}$, $7^{th}$ and $10^{th}$ layer). It can be seen that as the network deepens, the inattentive tokens are gradually removed or fused, while the most informative/attentive tokens are identified and preserved. In this way, the proposed method facilitates the ViTs in focusing on class-specific tokens in images. The visualization results also validate that our EViT is effective in dealing with images with either simple backgrounds or complex backgrounds.

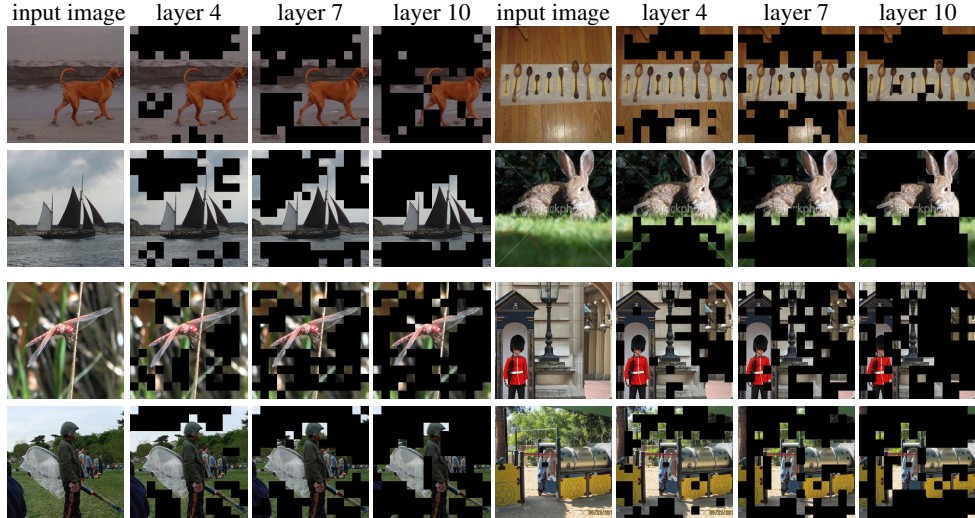

Figure 3: Visualization of inattentive tokens on EViT-DeiT-S with 12 layers. The masked regions represent the inattentive tokens that are fused into a new token. Our method can effectively identify inattentive tokens in images with either simple inattentive tokens (in the first two rows) or complex inattentive tokens (in the last two rows).

Table 2: Comparison on the two variants of EViT on DeiT-S (Touvron et al., 2021a). The values of the Top-1 and Top-5 accuracy are averaged over three independent trials. The number behind $\pm$ is the standard deviation of the three trials. The number in blue is the gap of the corresponding value w.r.t. the baseline DeiT-S.

| Keep rate | Top-1 Acc (%) | Top-5 Acc (%) | Throughput (images/s) | MACs | Top-1 Acc (%) | Top-5 Acc (%) | Throughput (images/s) | MACs |
|---|---|---|---|---|---|---|---|---|
| DeiT-S | 79.8 | 94.9 | 2923 | 4.6 | 79.8 | 94.9 | 2923 | 4.6 |
| | EViT without inattentive token fusion | | | | EViT with inattentive token fusion | | | |
| 0.9 | 79.9±0.1 (+0.1) | 94.9±0.0 (-0.0) | 3201 (+10%) | 4.0 (-13%) | 79.8±0.1 (-0.0) | 95.0±0.0 (+0.1) | 3197 (+9%) | 4.0 (-13%) |
| 0.8 | 79.7±0.0 (-0.1) | 94.8±0.0 (-0.1) | 3772 (+27%) | 3.5 (-24%) | 79.8±0.0 (-0.0) | 94.9±0.0 (-0.0) | 3619 (+24%) | 3.5 (-24%) |
| 0.7 | 79.4±0.1 (-0.4) | 94.7±0.1 (-0.2) | 4249 (+45%) | 3.0 (-35%) | 79.5±0.0 (-0.3) | 94.8±0.0 (-0.1) | 4385 (+50%) | 3.0 (-35%) |
| 0.6 | 79.1±0.2 (-0.7) | 94.5±0.1 (-0.4) | 4967 (+70%) | 2.6 (-43%) | 78.9±0.1 (-0.9) | 94.5±0.0 (-0.4) | 4722 (+62%) | 2.6 (-43%) |
| 0.5 | 78.4±0.2 (-1.4) | 94.1±0.0 (-0.8) | 5325 (+82%) | 2.3 (-50%) | 78.5±0.0 (-1.3) | 94.2±0.0 (-0.7) | 5408 (+85%) | 2.3 (-50%) |

## 4 EXPERIMENTS

**Implementation details.** We train all of the models on the ImageNet (Deng et al., 2009) training set with approximately 1.2 million images and report the accuracy on the 50k images in the test set. By default, the token identification module is incorporated into the $4^{th}$, $7^{th}$ and $10^{th}$ layer of DeiT-S and DeiT-B (with 12 layers in total) and incorporated into the $5^{th}$, $9^{th}$ and $13^{th}$ layer of LV-ViT-S (with 16 layers in total). The image resolution in training and testing is $224 \times 224$ unless otherwise specified. For the training strategies and optimization methods, we simply follow those in the original papers of DeiT (Touvron et al., 2021a) and LV-ViT (Jiang et al., 2021). Since our method can be easily incorporated into these models without making substantial modifications to them, the original training strategies work well with our method. Besides, we adopt a warmup strategy for attentive token identification. Specifically, the keep rate of attentive tokens is gradually reduced from 1 to the target value with a cosine schedule. Unlike DynamicViT (Rao et al., 2021), we do not use a pretrained ViT to initialize our models in most experiments, except in the experiments with an oracle ViT (see the following paragraphs). We train the models with EViT from scratch for 300 epochs on 16 NVIDIA A100 GPUs and measure the throughput of the models on a single A100 GPU with a batch size of 128 unless otherwise specified. The multiply-accumulate computations (MACs) metric is measured by `torchprofile` (Liu, 2021).

Table 3: Results of EViT on LV-ViT-S

| Keep rate | Top-1 Acc (%) | Top-5 Acc (%) | Throughput (images/s) | MACs (G) |
|---|---|---|---|---|
| LV-ViT-S | 83.3 | – | 2112 | 6.6 |
| 0.7 | 83.0 (-0.3) | 96.3 | 2954 (+40%) | 4.7 (-29%) |
| 0.5 | 82.5 (-0.8) | 96.2 | 3603 (+71%) | 3.9 (-41%) |

Table 4: Results of training EViT-DeiT-S with a keep rate of 0.7 for different epochs.

| Keep rate | Epochs | Top-1 Acc (%) | Top-5 Acc (%) | MACs (G) |
|---|---|---|---|---|
| 0.7 | 300 | 79.5 | 94.8 | 3.0 |
| 0.7 | 450 | 80.2 | 95.1 | 3.0 |
| 0.7 | 600 | 81.0 | 95.3 | 3.0 |

We report the main results of EViT on Tables 2 and 3. On both DeiT and LV-ViT, the proposed EViT achieves significant speedup while restricting the accuracy drop in a relatively small range. For example, DeiT-S trained with EViT with a keep rate of 0.7 increase the inference throughput by 50% while maintaining the Top-1 accuracy reduction within 0.3% on ImageNet.

**Inattentive token fusion.** To mitigate the problem of information loss in token removal, we propose inattentive token fusion, which fuses the non-topk tokens according to the attentiveness from the [CLS] token. We experimentally compare the proposed token reorganization method with and without inattentive token fusion. As shown in Table 2, token reorganization with inattentive token fusion typically outperforms the one without inattentive token fusion. Although the improvement is small, there is no additional computational overhead introduced. From another perspective, it indicates the effectiveness of our attentive token identification because the majority of effective tokens are well preserved. Moreover, the accuracy fluctuation in EViT with inattentive token fusion is smaller than the vanilla inattentive token removal, as shown by the standard deviation in Table 2. Inattentive token fusion also benefits EViT on high resolution images, as shown in Table 9.

**Epochs of training.** Since ViTs do not have inductive bias such as translational invariance processed by CNNs, they typically require more training data and/or training epochs to reach a comparable generalization performance as CNNs (Dosovitskiy et al., 2021; Touvron et al., 2021a). We find that training longer epochs continues to benefit ViTs in the efficient computation regime. We train the DeiT-S model with 0.7 keep rate for longer epochs of 450 and 600, respectively. As shown in Table 4, their performance steadily improves with training epochs.

**Training/Finetuning on higher resolution images.** By fusing the inattentive tokens, We are able to feed EViT with more tokens under the same computational cost. Therefore, we train and/or finetune EViT on resized images with higher resolutions than the standard resolution of $224^2$, and we report the results on Table 5. We can see that EViT in most tested cases performs favourably against a vanilla DeiT/LV-ViT, while having comparable or higher inference throughput than the baselines. Notably, training EViT-LV-ViT-S on images of resolution $224^2$ and further finetuning it on a higher resolution of $448^2$ for another 100 epochs gives a very competitive Vision Transformer model, achieving an ImageNet top-1 accuracy of 84.7%, which is 0.3% higher than LV-ViT-S@384 with basically the same throughput and number of parameters. The experimental results validate our hypothesis that images typically contain tokens that are less informative and contribute little to the recognition task. Since ViTs perform global self-attention among all tokens in as early as

Table 5: Results of training/finetuning on high resolution images. EViT-DeiT-S and EViT-LV-ViT-S have a throughput/MACs comparable to the baselines while achieving better recognition accuracy on ImageNet. The numbers before and after ↑ indicate the image size in the 300-epoch training and 100-epoch finetuning, respectively.

(a) DeiT-S

| Model | Keep rate | Image size | Top-1 (%) | Top-5 (%) | img/s | MACs (G) |
|---|---|---|---|---|---|---|
| DeiT-S | 1.0 | 224 | 79.8 | 94.9 | 2923 | 4.6 |
| EViT | 0.5 | 256 | 79.3 | 94.7 | 3788 | 3.1 |
| EViT | 0.5 | 288 | 80.1 | 95.0 | 3138 | 3.9 |
| EViT | 0.5 | 304 | 81.0 | 95.6 | 2905 | 4.4 |
| EViT | 0.6 | 256 | 80.0 | 95.0 | 3524 | 3.5 |
| EViT | 0.6 | 288 | 81.0 | 95.4 | 2927 | 4.5 |
| EViT | 0.7 | 272 | 80.3 | 95.3 | 2870 | 4.6 |

(b) LV-ViT-S

| Model | Keep rate | Image size | Top-1 (%) | Top-5 (%) | img/s | MACs (G) |
|---|---|---|---|---|---|---|
| LV-ViT-S | 1.0 | 224 | 83.3 | – | 2112 | 6.6 |
| LV-ViT-S | 1.0 | 224 ↑ 384 | 84.4 | – | 557 | 21.9 |
| EViT | 0.9 | 240 | 83.6 | 96.5 | 1956 | 6.8 |
| EViT | 0.8 | 256 | 83.6 | 96.6 | 1901 | 6.9 |
| EViT | 0.7 | 256 | 83.5 | 96.5 | 2102 | 6.2 |
| EViT | 0.7 | 272 | 83.7 | 96.6 | 1829 | 7.1 |
| EViT | 0.5 | 304 | 83.4 | 96.5 | 1758 | 7.4 |
| EViT | 0.7 | 272 ↑ 448 | 84.7 | 97.1 | 548 | 21.5 |

Table 6: Results of training EViT-DeiT-S and EViT-DeiT-B using DeiT-S as an oracle. Training for longer epochs continues to benefits the EViT in efficiency regime.

| Model | Oracle | Keep rate | Epochs | Top-1 (%) | Top-5 (%) | Throughput (img/s) | MACs (G) |
|---|---|---|---|---|---|---|---|
| EViT-DeiT-S | × | 0.7 | 300 | 79.5 | 94.8 | 4385 | 3.0 |
| EViT-DeiT-S | ✓ | 0.7 | 300 | 80.8 | 95.4 | 4385 | 3.0 |
| EViT-DeiT-S | ✓ | 0.7 | 450 | 81.0 | 95.5 | 4385 | 3.0 |
| EViT-DeiT-S | ✓ | 0.7 | 600 | 81.3 | 95.5 | 4385 | 3.0 |
| DeiT-B | × | 1.0 | 300 | 81.8 | 95.6 | 1295 | 17.6 |
| EViT-DeiT-B | × | 0.7 | 300 | 81.3 | 95.3 | 2053 | 11.6 |
| EViT-DeiT-B | ✓ | 0.7 | 300 | 82.1 | 95.6 | 2053 | 11.6 |

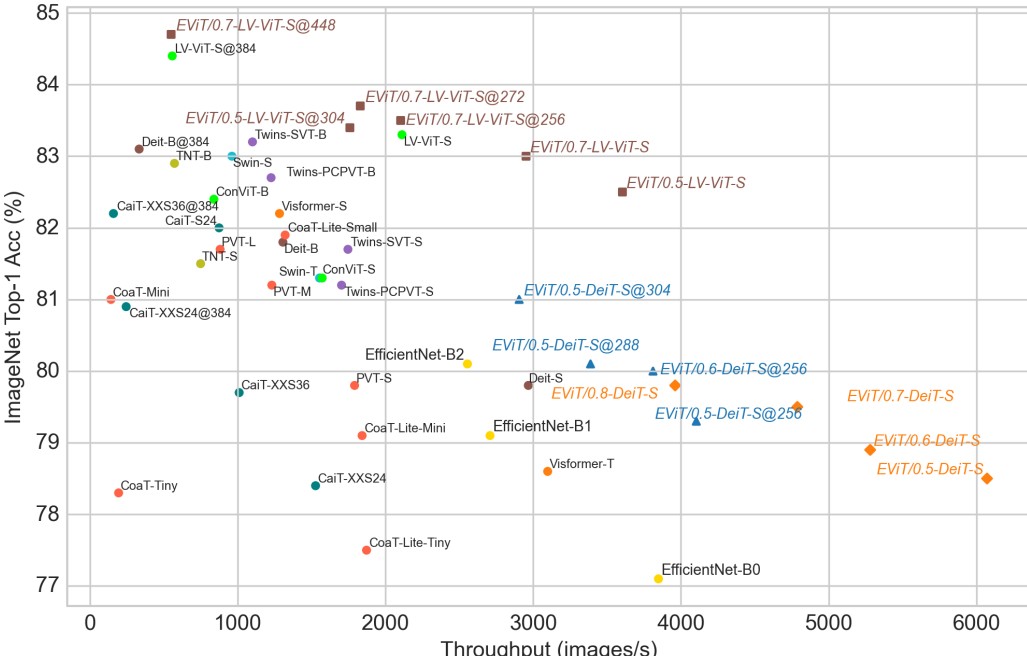

Figure 4: Comparison of different models with various accuracy-throughput trade-off. The proposed method EViT achieves better trade-off than the other methods (marked with circles). The throughput is measured on an NVIDIA A100 GPU using the largest possible batch size for each model. The input image size is $224^2$ unless specified after the @.

the first layer, the early interaction and information exchange between tokens makes it possible to discard/fuse some of the least informative tokens in intermediate ViT layers since they have been "seen" by other tokens (including the [CLS] token). In comparison, the receptive field of a CNN at the shallow layers is relatively small due to the locality property of convolution, which makes it difficult to reduce computation at early stages.

**Training with an oracle ViT.** In EViT, the criterion of selecting tokens is the attention between the [CLS] token and other tokens. Therefore, it would be very helpful if we know the importance of each token to the prediction tasks in advance. To this end, we introduce an oracle ViT to guide the [CLS] through the token selection process. A good oracle knows which tokens are important and which are not. For this purpose, we use a fully trained DeiT-S/B as an oracle and initialize EViT-DeiT-S/B with the parameters of the oracle ViT, such that the EViT know which tokens contribute more to the prediction result. We train EViT equipped with an oracle using the same training setup as training a vanilla EViT. As shown in Table 6, training with an oracle significantly improves the recognition accuracy of EViT.

Table 7: Comparisons on EViT and DynamicViT (Rao et al., 2021). ✓ indicates the model is initialized with a pre-trained DeiT-S. For fair comparison, the throughput is measured on the same machine with the same setting using a maximum batch size.

| Model | Pre-trained | Epochs | Keep rate | Top-1 (%) | #Params (M) | Throughput (img/s) | MACs (G) |
|---|---|---|---|---|---|---|---|
| DynamicViT-DeiT-S | ✓ | 30 | 0.5 | 77.5 | 22.8 | 5579 | 2.2 |
| EViT-DeiT-S | ✓ | 30 | 0.5 | 78.5 (+1.0) | 22.1 (-0.7) | 5549 | 2.3 |
| EViT-DeiT-S | ✓ | 100 | 0.5 | 79.1 (+1.6) | 22.1 (-0.7) | 5549 | 2.3 |
| DynamicViT-DeiT-S | ✓ | 30 | 0.7 | 79.3 | 22.8 | 4439 | 3.0 |
| EViT-DeiT-S | ✓ | 30 | 0.7 | 79.5 (+0.2) | 22.1 (-0.7) | 4478 | 3.0 |
| EViT-DeiT-S | ✓ | 100 | 0.7 | 79.8 (+0.5) | 22.1 (-0.7) | 4478 | 3.0 |
| DynamicViT-DeiT-S | ✕ | 300 | 0.5 | 73.1 | 22.8 | 5579 | 2.2 |
| EViT-DeiT-S | ✕ | 300 | 0.5 | 78.5 (+5.4) | 22.1 (-0.7) | 5549 | 2.3 |
| DynamicViT-DeiT-S | ✕ | 300 | 0.7 | 77.6 | 22.8 | 4439 | 3.0 |
| EViT-DeiT-S | ✕ | 300 | 0.7 | 79.5 (+1.9) | 22.1 (-0.7) | 4478 | 3.0 |

**Comparison with DynamicViT.** We compare the proposed EViT with DynamicViT (Rao et al., 2021). Note that the authors of DynamicViT used a finetuning strategy. Therefore, we first compare the two methods in finetuning, and then we also train DynamicViT from scratch for 300 epochs to compare it with our baselines. In the case of training from scratch, no pretrained ViT can be used, and thus the distillation loss and KL loss in DynamicViT are discarded. The results in Table 7 show that EViT outperforms DynamicViT in recognition accuracy under the same computational cost while EViT uses fewer parameters. Moreover, the accuracy of EViT continues to improve in longer-epoch training. When training from scratch, a significant accuracy reduction is observed in DynamicViT, which suggests that DynamicViT requires a teacher ViT (i.e., the parameter initialization, the distillation loss, and the prediction (KL) loss from a pretrained ViT (Rao et al., 2021)) to obtain a good performance.

**Comparison with other Vision Transformers.** We plot the accuracy and throughput of various ViTs in Figure 4, which shows EViT is very competitive among the vision models in terms of computation-accuracy trade-off. For EViT, we only plot the EViT models that are trained from scratch for fair comparisons, except EViT/0.7-LV-ViT-S@448, which is finetuned for 100 epochs. The comparing methods include DeiT (Touvron et al., 2021a), CaiT (Touvron et al., 2021b), LV-ViT (Jiang et al., 2021), CoaT (Xu et al., 2021), Swin (Liu et al., 2021), Twins (Chu et al., 2021), Visformer (Chen et al., 2021), ConViT (d'Ascoli et al., 2021), TNT (Han et al., 2021a), Efficient-Net (Tan & Le, 2019), and PVT (Wang et al., 2021a). These ViTs variants focus on modifying the ViT architectures or the interaction methods between image tokens to achieve improvement over the vanilla ViT (Dosovitskiy et al., 2021). Thus, their methods and ours are complementary to each other, which makes it possible to incorporate our method into some of the ViT variants.

## 5 CONCLUSION

In this paper, we present a token reorganization method. By identifying the tokens with the largest attention from the class token, the proposed EViT reaches a better trade-off between accuracy and efficiency than various Vision Transformer models. Moreover, we propose inattentive token fusion, which fuses the information from less informative tokens to a new token. Inattentive token fusion improves both the recognition accuracy and training stability. Experimentally, we apply the proposed token reorganization method to two variants of Vision Transformer, namely, DeiT (Touvron et al., 2021a) and LV-ViT (Jiang et al., 2021). In both variants, EViT achieves a significant speedup in inference while the reduction in recognition accuracy is relatively small. Moreover, when training on higher resolution images, EViT improves the accuracy to various extents while maintaining a similar or smaller computation cost as the original DeiT/LV-ViT models. Besides, when equipped with an oracle ViT which knows which tokens are more important, EViT can achieve further improvement on the trade-off between accuracy and efficiency. The proposed EViT can be easily adapted in ViTs and brings no additional parameters, nor does it require sophisticated training strategies. The proposed token reorganization method can serve as an effective acceleration approach for Vision Transformers.

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

# A VISUALIZATION

In this part, we present more visualization results in Figure 5 to show the attentive token identification. The input images are randomly selected from the ImageNet dataset. The results validate that our EViT is able to deal with different images from various categories.

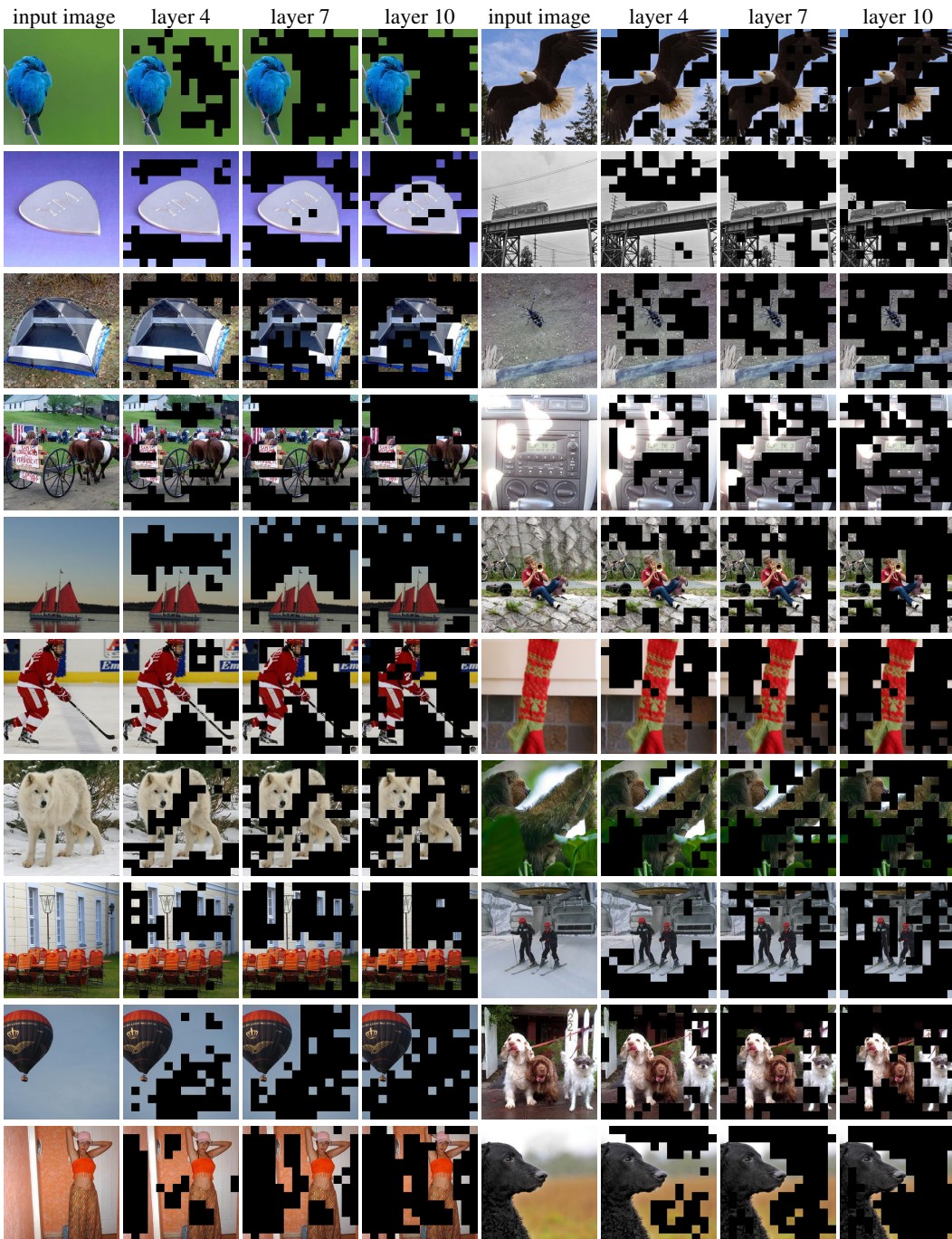

Figure 5: Extended visualization results of inattentive tokens on EViT-DeiT-S with 12 layers.. The regions without masks represent the attentive tokens. The masked regions denote the inattentive tokens that are fused into a new token. Our EViT is effective in dealing with images from different categories.

**Algorithm 1:** PyTorch-like pseudocode of EViT for a ViT encoder.

```
# H: number of attention heads
# N: number of input tokens
# C: the dimension of token vector
# k: the token keep rate
# x: the input tokens with shape [N, C], with the first being the [CLS]
    token
# fc_q, fc_k, fc_v: linear transforms for query, key, and value of self-
    attention
# @: matrix multiplication
# proj: linear projection in self-attention
# norm: layer normalization
# ffn: feed-forward network

# initialize
avg_cls_attn = zeros(N-1)
x_out = []
x_residual = x
x = norm(x)

## multi-head self-attention computation
for i in range(0, H): # compute self-attention for each attention head
    # linearly map the tokens to query, key, and value matrices
    q, k, v = fc_q[i](x), fc_k[i](x), fc_v[i](x)
    # compute the attention map
    attn = (q @ k.transpose()) / sqrt(C/H)
    attn = softmax(attn, dim=1)

    x_head = attn @ v
    x_out.append(x_head)

    # compute the [CLS] attentiveness w.r.t. other tokens,
    # without using [CLS] attention to itself
    cls_attn = attn[0, 1:]
    avg_cls_attn += cls_attn

# concatenate the output tokens of all heads
x = concat(x_out, dim=1)

x = proj(x) # shape: [N, C]
x = x + x_residual

# average the [CLS] attentiveness over all heads
avg_cls_attn /= H

# sort the avg_cls_attn in descending order
sorted_cls_attn, idx = sort(avg_cls_attn)

# compute the number of attentive tokens, without counting the [CLS] token
K = ceil(k * (N - 1))

topk_attn, topk_idx = sorted_cls_attn[:K], idx[:K]
non_topk_attn, non_topk_idx = sorted_cls_attn[K:], idx[K:]

# separate [CLS] token and other tokens
cls_token = x[0:1]
x_without_cls = x[1:]

# obtain the attentive and inattentive tokens
attentive_tokens = x_without_cls[topk_idx]
inattentive_tokens = x_without_cls[non_topk_idx]

# compute the weighted combination of inattentive tokens
fused_token = non_topk_attn @ inattentive_tokens

# concatenate these tokens
x_new = concat([cls_token, attentive_tokens, fused_token], dim=0)

x_residual = x_new
x_new = norm(x_new)
x_new = ffn(x_new)
x_new = x_new + x_residual
return x_new
```

Table 8: Results of EViT on DeiT-B

| Keep rate | Top-1 Acc (%) | Top-5 Acc (%) | Throughput (images/s) | MACs (G) |
|-----------|---------------|---------------|-----------------------|----------|
| DeiT-B | 81.8 | 95.6 | 1295 | 17.6 |
| 0.9 | 81.8 (-0.0) | 95.6 (-0.0) | 1441 (+11%) | 15.3 (-13%) |
| 0.8 | 81.7 (-0.1) | 95.4 (-0.2) | 1637 (+26%) | 13.2 (-25%) |
| 0.7 | 81.3 (-0.5) | 95.3 (-0.3) | 2053 (+59%) | 11.5 (-35%) |
| 0.6 | 80.9 (-0.9) | 95.1 (-0.5) | 2177 (+68%) | 10.0 (-43%) |
| 0.5 | 80.0 (-1.8) | 94.5 (-1.1) | 2482 (+92%) | 8.7 (-51%) |

Table 9: Comparisons of the two variants (i.e., vanilla inattentive token removal and inattentive token fusion) of EViT-DeiT-S on higher resolution images.

| Method | Keep rate | Image size | Top-1 Acc (%) | Top-5 Acc (%) | Throughput (images/s) | MACs (G) |
|--------|-----------|------------|---------------|---------------|-----------------------|----------|
| DeiT-S | 1.0 | 224 | 79.8 | 94.9 | 2923 | 4.6 |
| Inattentive token removal | 0.5 | 304 | 80.9 (+1.1) | 95.3 (+0.4) | 2927 | 4.4 |
| Inattentive token fusion | 0.5 | 304 | 81.0 (+1.2) | 95.6 (+0.7) | 2905 | 4.4 |

## B  EXTENDED EXPERIMENTS

**More results of EViT on DeiT.** We provide the performance of EViT on DeiT-B (Touvron et al., 2021a) in Table 8. Again, the resulting models of EViT-DeiT-B achieve significant speedup while restricting the accuracy drop in a relatively small range. For example, EViT-DeiT-B with a keep rate of 0.8 increases inference throughput by 26% while maintaining the Top-1 accuracy reduction within 0.1% on ImageNet.

**More comparisons of vanilla inattentive token removal and inattentive token fusion.** We compare the performance of the two variants of EViT, namely, vanilla inattentive token removal and inattentive token fusion, on the training of higher resolution images. In this scenario, EViT with inattentive token fusion also outperforms the EViT with vanilla inattentive token removal, especially in the Top-5 accuracy, as shown in Table 9. Although the improvement is relatively small, it is practical as inattentive token fusion brings no additional computational overhead compared to vanilla inattentive token removal.

**Using tokens-to-tokens attentive scores for inattentive tokens identification.** As discussed in Section 3.2, we use the attention from the `[CLS]` token to other tokens as the attentive scores to identify the class-related tokens that are important to visual recognition. This attentive/inattentive token identification approach is motivated by the fact that the final prediction of ViTs is determined by the representation of the `[CLS]` token in the last layer. However, we also find that it is possible to use a tokens-to-tokens attention approach to identify the attentive tokens, where the attention scores are computed as the average of the attention from all tokens to all tokens. Specifically, recall the attention map $A$ is computed by:

$$A = \text{Softmax}(\frac{QK^\top}{\sqrt{d}}). \tag{3}$$

The attentive score is computed by

$$a = \frac{1}{n}\sum_{i=1}^{n} A_{i,:} \tag{4}$$

where $n$ is the number of tokens and $A_{i,:}$ is the $i$-th row of the attention map $A$. When there are multiple self-attention heads, the final attentive score is averaged over all heads: $\bar{a} = \sum_{h=1}^{H} a^{(h)}/H$, where $H$ is the number of heads. In a nutshell, in the tokens-to-tokens strategy, each token can "vote" to decide which tokens are attentive, while in the `[CLS]`-to-tokens strategy, only the `[CLS]` token can "vote". As shown in Table 10, EViT with the tokens-to-tokens attention strategy performs comparably with the `[CLS]`-to-tokens attention strategy but is less efficient. Nevertheless, this experimental observation opens the door for incorporating the proposed EViT into the ViT variants with no `[CLS]` tokens (e.g., PVT (Wang et al., 2021a)).

Table 10: The performance of EViT-DeiT-S with different attentive token identification strategies.

| Method | Inattentive token fusion | Keep rate | Top-1 Acc (%) | Top-5 Acc (%) | Throughput (img/s) | MACs (G) |
|---|---|---|---|---|---|---|
| [CLS]-to-tokens attentiveness | × | 0.5 | 78.4 | 94.1 | 5325 | 2.3 |
| | ✓ | 0.5 | 78.5 | 94.2 | 5408 | 2.3 |
| | × | 0.7 | 79.4 | 94.7 | 4249 | 3.0 |
| | ✓ | 0.7 | 79.5 | 94.8 | 4385 | 3.0 |
| Tokens-to-tokens attentiveness | × | 0.5 | 78.4 | 94.2 | 5241 | 2.3 |
| | ✓ | 0.5 | 78.5 | 94.3 | 5346 | 2.3 |
| | × | 0.7 | 79.4 | 94.8 | 4152 | 3.0 |
| | ✓ | 0.7 | 79.4 | 94.7 | 4290 | 3.0 |

Table 11: Training time for 300-epoch training of DeiT-S and EViT-DeiT-S with different keep rates.

| Keep rate | Top-1 Acc (%) | Top-5 Acc (%) | Throughput (images/s) | MACs (G) | Training time (GPU×hours) |
|---|---|---|---|---|---|
| DeiT-S | 79.8 | 94.9 | 2923 | 4.6 | 201 |
| 0.9 | 79.8 (-0.0) | 95.0 (+0.1) | 3197 (+9%) | 4.0 (-13%) | 188 (-6%) |
| 0.8 | 79.8 (-0.0) | 94.9 (-0.0) | 3619 (+24%) | 3.5 (-24%) | 174 (-13%) |
| 0.7 | 79.5 (-0.3) | 94.8 (-0.1) | 4385 (+50%) | 3.0 (-35%) | 160 (-20%) |
| 0.6 | 78.9 (-0.9) | 94.5 (-0.4) | 4722 (+62%) | 2.6 (-43%) | 150 (-25%) |
| 0.5 | 78.5 (-1.3) | 94.2 (-0.7) | 5408 (+85%) | 2.3 (-50%) | 145 (-28%) |

**Training time of EViT.** We provide a comparison on the training time of DeiT-S (Touvron et al., 2021a) and EViT-DeiT-S in Table 11. The results show that EViT requires considerably smaller GPU×hours to train when compared to a vanilla DeiT under the same number of epochs. Since EViT continues to improve in recognition accuracy with larger training epoch (as shown in Table 4), the performance gap between a vanilla ViT and an EViT can be narrowed down and even closed when the EViT is trained for the same amount of time as the ViT.

**Finetuning of EViT on high resolution images.** We provide results of finetuning EViT on high-resolution images in the last three rows in Table 12. To finetune EViT-DeiT-S from low-resolution images to high-resolution images, we need to resolve the discrepancy between the positional embedding (Touvron et al., 2021a) of two different resolutions. To this end, we interpolate the positional embedding of lower resolution to the desired higher resolution. This method works quite well in our experiments, as shown in the last three rows in Table 12. However, directly training on higher resolution images seems a better way to train EViT, as it requires less training time (because no finetuning is performed) and typically obtains higher accuracy than finetuning.

**Token reorganization locations.** It is possible that different combinations of keep rates and token reorganization locations can reach the same computational efficiency. For example, we can a) move the reorganization modules into shallower layers and increase the keep rates, or b) move the reorganization modules into deeper layers and decrease the keep rates, to keep the computation cost (approximately) unchanged. To get a deeper understanding of this aspect, we train EViT-DeiT-S models with different token reorganization locations and keep rates, each of which has a similar computational cost as the baseline. The results in Table 13 reveal two perspectives. First, moving the reorganization modules into shallower layers deteriorates the accuracy. For example, when the token reorganization module is placed before the third layer (i.e., at the first or second layer), the recognition accuracy drops considerably even though the computational cost is the same. This suggests that ViT cannot identify the important tokens at the early stages, which is quite reasonable as the processing of input tokens is just started at shallow layers, where the attention maps are unreliable for token removal. Second, placing the reorganization modules in different deeper layers has only marginal influence on the accuracy. For instance, when the token reorganization modules are placed behind the third layer, the resulting models have basically the same accuracy. This suggests that EViT has stable performance as long as the reorganization locations are not in shallow layers. Because of this reason, we refrain from searching for a (slightly) better configuration of the reorga-

Table 12: Results of training/finetuning on high resolution images. The numbers before and after ↑ indicate the image size in the 300-epoch training and 50-epoch finetuning, respectively.

| Model | Keep rate | Image size | Top-1 (%) | Top-5 (%) | Throughput (img/s) | MACs (G) |
|-------|-----------|------------|-----------|-----------|--------------------|----------|
| DeiT-S | 1.0 | 224 | 79.8 | 94.9 | 2923 | 4.6 |
| EViT-DeiT-S | 0.5 | 256 | 79.3 | 94.7 | 3788 | 3.1 |
| EViT-DeiT-S | 0.5 | 288 | 80.1 | 95.0 | 3138 | 3.9 |
| EViT-DeiT-S | 0.5 | 304 | 81.0 | 95.6 | 2905 | 4.4 |
| EViT-DeiT-S | 0.6 | 256 | 80.0 | 95.0 | 3524 | 3.5 |
| EViT-DeiT-S | 0.6 | 288 | 81.0 | 95.4 | 2927 | 4.5 |
| EViT-DeiT-S | 0.7 | 272 | 80.3 | 95.3 | 2870 | 4.6 |
| EViT-DeiT-S | 0.5 | 224 ↑ 304 | 80.3 | 95.2 | 2905 | 4.4 |
| EViT-DeiT-S | 0.6 | 224 ↑ 288 | 80.7 | 95.3 | 2927 | 4.5 |
| EViT-DeiT-S | 0.7 | 224 ↑ 272 | 80.6 | 95.2 | 2870 | 4.6 |

Table 13: The performances of EViT-DeiT-S with different of keep rates and reorganization locations (R.L.). For fair comparison, we keep the variants having the same level of computational cost (i.e., MACs) as our EViT baseline in the first row by tuning the keep rates.

| #R.L. | Reorganization locations | Keep rates | Top-1 Acc (%) | Top-5 Acc (%) | MACs (G) |
|-------|--------------------------|------------|---------------|---------------|----------|
| 3 | $[4, 7, 10]$ | 0.7 | 79.50 | 94.77 | 3.0 |
| 3 | $[5, 7, 10]$ | $[0.64, 0.70, 0.70]$ | 79.57 (+0.07) | 94.80 | 3.0 |
| 3 | $[3, 7, 10]$ | $[0.74, 0.70, 0.70]$ | 79.47 (-0.03) | 94.72 | 3.0 |
| 3 | $[4, 8, 10]$ | $[0.70, 0.64, 0.70]$ | 79.64 (+0.14) | 94.86 | 3.0 |
| 3 | $[4, 6, 10]$ | $[0.70, 0.75, 0.70]$ | 79.54 (+0.04) | 94.78 | 3.0 |
| 3 | $[6, 8, 10]$ | 0.6 | 79.43 (-0.07) | 94.69 | 3.0 |
| 3 | $[2, 6, 10]$ | 0.765 | 78.72 (-0.78) | 94.22 | 3.0 |
| 3 | $[2, 4, \ \ 6]$ | 0.81 | 78.45 (-1.05) | 94.26 | 3.0 |
| 3 | $[1, 5, \ \ 9]$ | $[0.80, 0.79, 0.79]$ | 76.34 (-3.16) | 93.14 | 3.0 |
| 4 | $[4, 6, 8, 10]$ | 0.76 | 79.38 (-0.12) | 94.72 | 3.0 |
| 4 | $[2, 5, 8, 11]$ | $[0.90, 0.77, 0.62, 0.50]$ | 78.93 (-0.57) | 94.52 | 3.0 |
| 7 | $[4, 5, 6, 7, 8, 9, 10]$ | 0.85 | 79.39 (-0.11) | 94.68 | 3.0 |

nization locations and keep rates. We adopt a simple strategy to decide the reorganization locations in our experiments, where the token reorganization locations cut the ViT into blocks with the same number of layers. Specifically, for a ViT with $L$ layers and $t$ token reorganization layers in total, we first obtain the separating length $s = L/(t + 1)$. Then, the layer indices of the reorganization layers are $[s + 1, 2s + 1, \ldots, ts + 1]$, which cuts the ViT evenly. For the keep rates, we simply set them to the same value for each token reorganization module in the EViT.

**Adopting the DINO attention map for inattentive token identification.** In DINO (Caron et al., 2021), the authors showed that the [CLS] attention produced by a self-supervised trained ViT is naturally focused on the object parts in images. Therefore, one may wonder if the good DINO attention can be used in EViT for inattentive token identification. To answer this question, we use the [CLS] attention produced by the last layer of DINO to guide the token selection process in EViT. Instead of starting the token reorganization process at the $4^{th}$ layer of EViT-DeiT-S as we have done in previous experiments, we start the token reorganization at the first layer. Specifically, we select the top 50% tokens that correspond to the positions with the strongest DINO attention (i.e., corresponding to the top 50% attention values in the DINO attention). Experimental results in Table 14 show that EViT works well with the DINO attention. As a comparison, when the reorganization layer is placed in the first layer in a vanilla EViT, as shown in Table 13, a significant accuracy reduction is observed, while EViT equipped with DINO can maintain its recognition ability even if 50% of tokens are removed as early as the first layer. However, this is not an efficient approach in practice because obtaining the DINO attention requires forwarding the input through

Table 14: The performance of EViT-DeiT-S equipped with the DINO attention. The "MACs" in the table is the computational cost of the EViT-DeiT-S model and "Combined MACs" is the sum of the MACs of both EViT-DeiT-S and the DINO model (i.e., a DeiT-S) which is used to obtain the DINO attention.

| Method | Keep rate | Drop layer | Top-1 Acc (%) | Top-5 Acc (%) | MACs (G) | Combined MACs (G) |
|---|---|---|---|---|---|---|
| DINO-EViT-DeiT-S | 0.5 | $1^{st}$ | 78.7 | 94.2 | 2.2 | 6.8 |

Table 15: An ablation of token removal strategy in EViT-DeiT-S. EViT-topk is the method of keeping the tokens with the largest [CLS]-to-token attention values. EViT-random denotes the method of randomly keeping the tokens, while EViT-mink is the method of keeping the tokens with the smallest [CLS]-to-token attention values.

| Method | Keep rate | Top-1 Acc (%) | Top-5 Acc (%) | MACs (G) |
|---|---|---|---|---|
| EViT-topk | 0.5 | 78.5 | 94.2 | 2.3 |
| EViT-random | 0.5 | 77.3 (-1.2) | 93.5 (-0.7) | 2.3 |
| EViT-mink | 0.5 | 75.2 (-3.3) | 92.4 (-1.8) | 2.3 |
| EViT-topk | 0.7 | 79.5 | 94.8 | 3.0 |
| EViT-random | 0.7 | 78.4 (-1.1) | 94.2 (-0.6) | 3.0 |
| EViT-mink | 0.7 | 76.4 (-3.1) | 93.2 (-1.6) | 3.0 |

the DINO model (i.e., a DeiT model) and bringing extra computational burden, as shown by the "Combined MACs" in Table 14.

**The correlation between the quality of the attention masks and the performance of EViT.** From Figures 3 and 5, we can see that the masks produced by EViT are quite intuitive and of good quality, which means that the attention values of the CLS token to other tokens correlate well with the semantic segmentation of images. Therefore, we can deliberately use low quality masks in EViT to see how its performance changes with the quality of the masks. Note that we use the largest [CLS]-to-tokens attention values to obtain the mask for the keeping tokens (we call this method EViT-topk). Thus, the worse possible mask one can obtain is probably using the smallest [CLS]-to-tokens attention values (we call this method EViT-mink), and a not-so-bad mask can be obtained by randomly generating a mask without considering the [CLS]-to-tokens attention (we call this method EViT-random). We perform experiments to see the classification accuracy of EViT under these circumstances. As shown in Table 15, the classification accuracy of these three methods clearly correlates with the quality of the masks. From the experimental results, we conclude that masks of better quality help improve the accuracy. Therefore, we believe it is a potential direction for future research as to how to (efficiently) obtain a better mask to improve the performance of EViT.

**The attention scores of the inattentive tokens w.r.t. the layer depth.** In order to find how the attentive scores of the inattentive tokens evolve, we randomly sample several images and plot the change of the [CLS]-to-tokens attention values of $n$ inattentive tokens ($n = 25$) w.r.t. the depth of layers in Fig. 6. Specifically, we use a pre-trained DeiT-S and identify the $n$ tokens with the smallest [CLS]-to-tokens attention in the $4^{th}$ layer. Then we keep track of these selected tokens in the subsequent layers and draw the box plot to see how the statistics (e.g., the mean, standard deviation, and maximum and minimum values) change with the layer depth. The results suggest that the tokens that are identified as inattentive tokens in the shallow layer tend to be less preferred in deeper layers.

**The evolution of the inattentive tokens during EViT training process.** To prove the effectiveness of incorporating EViT into the training process of ViT, we present the evolution of the inattentive tokens during the training of EViT-DeiT-S with a keep rate of 0.7 in Figure 7. It can be observed that at the initial training stage, the EViT model tends to misidentify the inattentive tokens (e.g., the bulk of the car in the second column of the first three rows). However, the inattentive tokens gradually converge to the less informative regions of images during the training process. For example, in the last column in Figure 7, the masked regions mainly focus on the background (e.g., the grass). This observation validates the effectiveness of our proposed token identification methods.

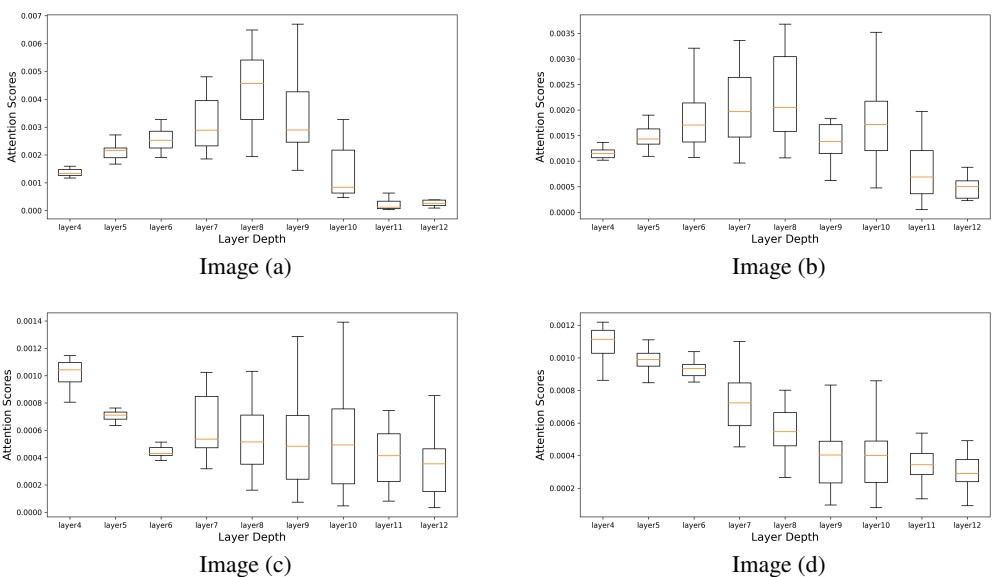

Figure 6: The box plot of the evolution of the attention scores of the inattentive tokens with the layer depth. The orange line in the box plot indicates the averaged attention scores of the inattentive tokens at a certain layer.

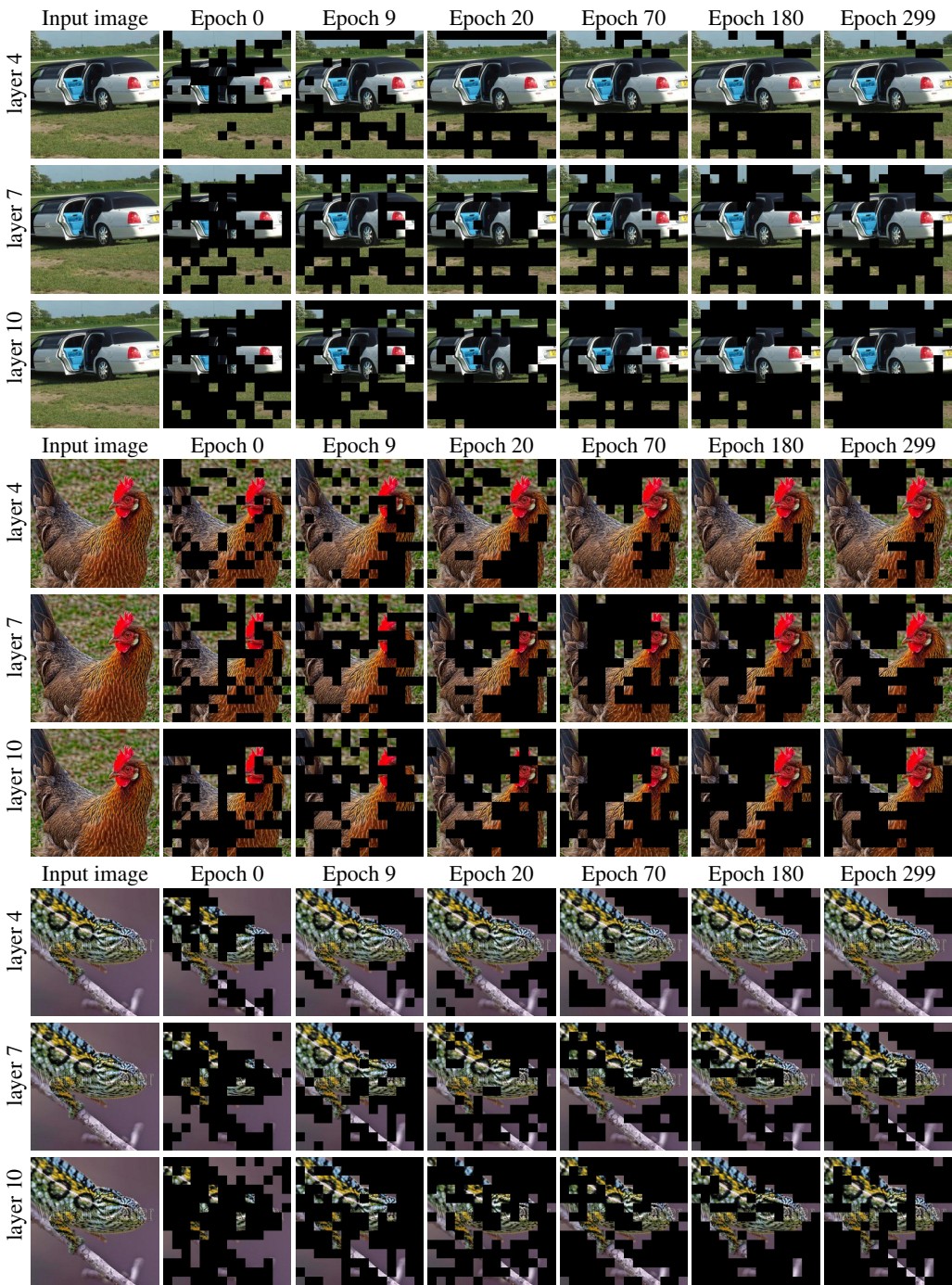

Figure 7: Visualization results of the evolution of the removed tokens during the EViT training process at different layers. The regions without masks represent the attentive tokens. The masked regions denote the inattentive tokens that are fused into a new token. At the initial training stage (e.g., the first epoch), the model is not stable in identifying the attentive tokens. As the training proceeds, EViT gradually converges, producing meaningful masks.

