# OpenReview forum: "EViT: Expediting Vision Transformers via Token Reorganizations"
_ICLR.cc/2022/Conference — ICLR 2022 Spotlight_

### Official Review · Reviewer_soMj · 2021-10-28

**Correctness:** 3
**Technical Novelty And Significance:** 4
**Empirical Novelty And Significance:** 4
**Recommendation:** 8
**Confidence:** 5

**Main Review:**

Pros:

- The idea of this  paper is interesting. Different from the previous DynamicViT that reduces tokens with an extra sub-network,
this work starts from the working principle of the self-attention module and leverages the relationships between the class token and the normal image tokens. I believe this paper provides a promising way to reduce the computational cost of vision transformers for future research.

- The motivation and presentation of this paper are clear. Pseudo code is provided. It is easy to follow for readers. I am looking forward to the  release of the whole code.

- Experiments are thorough. Results on both DeiT and LV-ViT show that the proposed approach can efficiently compress the computational cost while maintain the baseline results.

Cons:

- One of the issue that I concern is the performance of the inattentive token fusion. As shown in Table 2, it seems that the classification performance gap between the two sub-tables is rather small. The question is why the authors introduce the inattentive token fusion?
In addition, as described in the late part of the experiment section, training on images with higher resolutions can improve the model performance. So, the question is how the inattentive token fusion would affect the model performance when taking images with larger image sizes as input?

- In Page 7, the authors report that training with a longer schedule benefits the model performance when utilizing the proposed approach. Is there any evidence/observation showing that what leads to such a phenomenon?

- In the experiments, three reorganization layers are used. Have you attempted to use more?

- The proposed approach can be regarded as a type of downsampling operations. So, would the proposed approach perform better than previous classic downsampling operations, such as pooling and convolution. A typical counterpart should be the PVT model.

- In Fig. 4, the authors show the comparison of the proposed method with other vision transformers, but there is no corresponding content describing the differences between the proposed approach and other methods. It would be great to add a new paragraph to give more description.

Typos:   In Page 7, 'wee' should be 'we.' In Page 9, 'reach' should be 'reaches.' Please carefully revised the manuscript and correct the typos.



**Summary Of The Paper:**

This paper aims to expediting vision transformers by reducing the number of tokens. The main contribution is  the attentive token identification, which is based on calculating the attentiveness of the class token with respect to each image token. Experiments on DeiT and LV-ViT show that the proposed approach is able to successfully reduce the computational cost of the baseline models with nearly no performance drop.

**Summary Of The Review:**

In spite of some experiments missing, regarding the significance of the novolty of this paper, I think it deserves a positive score. If the concerns can be well addressed, I would like to lift the rating.

---

> ### Author Response · Authors · 2021-11-21
> **Response to Reviewer soMj**
>
> We would like to thank the reviewer for providing valuable comments and we answer the raised questions below.
>
> **1. The release of the whole code.**
>
> We will release the whole code of EViT to ease related research.
>
>
> **2. The performance of the inattentive token fusion.**
>
> We agree that the improvement upon using inattentive token fusion is small. However, the improvement is not affected by training noise. We report our ablation results in Table 2 where we train EViT three times independently and average the evaluation results. We have also included the standard deviation of these three trials in Table 2. It shows that when compared to a vanilla model, there is a lower accuracy fluctuation by using inattentive token fusion.
>
> Our intuition of introducing inattentive token fusion is to recover the dropped tokens which are still able to contribute to the prediction results. Although the improvement is small, there is no additional computational overhead introduced. From another perspective, it indicates the effectiveness of our attentive token identification because the majority of effective tokens are well preserved.
>
>
> **3. Performance with a longer training schedule.**
>
> The phenomenon of training with a longer schedule benefits EViT is a characteristic of ViTs. As has been studied in previous work [a,b], ViTs need massive data and longer training schedules to fully explore their potential, which is probably due to the lack of inductive bias in Vision Transformers [a,b]. As our token reduction is inherently integrated into ViTs, a longer training schedule benefits our EViT as well.
>
>
> **4. Performance with more reorganization layers.**
>
> We have conducted more experiments on the reorganization layers (including locations and numbers) and the tokens keep rates. The results of using four and seven reorganization layers are shown in Table 13 in the appendix, which reveals that as long as the reorganization locations are not in shallow layers, using more reorganization layers has little influence on the recognition accuracy under the same computational cost. On the other hand, if the reorganization layers are placed in shallow layers of the ViT (i.e., in the first or second layer), the resulting accuracy drops considerably regardless of the number of reorganization layers used.
>
>
>
> **5. Comparisons with classic downsampling operations.**
>
> As suggested, we have plotted the accuracy-computation trade-off of three PVT models (denoted by PVT-S, PVT-M, and PVT-L) in Figure 4 in the revised manuscript. Under the same computational cost, our method achieves higher accuracy than PVTs. An important difference between EViT and classic downsampling operations is that EViT dynamically downsample images according to the attentiveness of the image patches, while classic methods such as pooling and convolution typically perform downsampling on each local region of an image without considering the semantic richness of the image patches, which makes them less effective in the efficiency regime.
>
>
> **6. Comparisons with other methods in Figure 4.**
>
> We have provided a description of the differences between our method and the methods shown in Figure 4 in the revised manuscript. To summarize, the methods in Figure 4 have different motivations. They focus on modifying the ViT architectures or the interaction methods between image tokens to improve the ViTs, while we focus on the removal of less informative tokens. From this perspective, their methods and ours are complementary to each other and our method may be incorporated into some of them. For simplicity, we have only chosen two of the most popular ViTs variants, namely, DeiT and LV-ViT, to demonstrate the effectiveness of our approach.
>
>
> **7. Typos**
>
> Thank you for pointing out the typos. We have carefully revised the manuscript and corrected all the typos.
>
>
> **Reference**
>
> [a]. Alexey Dosovitskiy, Lucas Beyer, Alexander Kolesnikov, DirkWeissenborn, Xiaohua Zhai, Thomas Unterthiner, Mostafa Dehghani, Matthias Minderer, Georg Heigold, Sylvain Gelly, et al. An image is worth 16x16 words: Transformers for image recognition at scale. In ICLR, 2021.
>
> [b]. Hugo Touvron, Matthieu Cord, Matthijs Douze, Francisco Massa, Alexandre Sablayrolles, and Herve Jegou. Training data-efficient image transformers & distillation through attention. In ICML, 2021a.

---

> > ### Comment · Reviewer_soMj · 2021-11-30
> > **Rating**
> >
> > I would like to thank the reviewers for the reponses. Most of my concerns have been solved. Based on the quality of this paper and the reviews from other reviewers, I would like to lift the score from 6 to 8.

---

### Official Review · Reviewer_WtKh · 2021-10-29

**Correctness:** 3
**Technical Novelty And Significance:** 4
**Empirical Novelty And Significance:** 3
**Recommendation:** 8
**Confidence:** 5

**Main Review:**

## Strengths
1. The paper is well written and easy to follow.
2. The method is simple and does not require a pre-trained model.
3. The visualization looks good and intuitive.
4. ViT models are accelerated by a large margin.

## Neutral
1. Marginal improvements over the baselines with the standard resolution and epochs. The gain is more significant with more input tokens and a longer training schedule. I think this is acceptable as the method reduces the computation cost for each iteration.

## Main Weaknesses
1. It is arguably difficult to combine the proposed method with multi-scale transformers, e.g. Swin transformer. It might be doable but the real throughput gain could be very limited. The current performance improvement on the vanilla ViT cannot cover this limitation.
2. The ablation difference of inattentive token fusion is very limited and close to pure training noise. Thus, the discussion of the vanilla model’s training instability and the improved training efficiency lacks enough justification.
3. It would be interesting to see if one can directly train DynamicViT from scratch. There seems no reason prevents us from doing so. As this is one of the main advantages of the proposed method over DynamicViT, more discussion is expected.
4. Apart from the class token, one could also use the full tokens-to-tokens attention to compute the attentive score. This is an obvious alternative but is not sufficiently studied.

## Suggestions
1. It is interesting to see if we can make use of the good DINO mask as an attentive region, although the DINO mask requires much longer training.
2. The interpretability of the learned mask could be evaluated just like what is done in DINO and then we may be able to see how the mask quality correlates with performance.
3. To better support the intuition of dropping tokens, one could visualize (in a naively pre-trained ViT) whether a token is never attended by the class token once it is less preferred (i.e. roughly monotonically decreasing attention from the class token). If there are cases where a token is only attended in the last layer, then the intuition of dropping-a-token-forever may not hold.


**Summary Of The Paper:**

This paper proposes a mechanism to dynamically drop transformer tokens, according to class token attention score. It is able to reduce computation with similar accuracy or improve accuracy with controlled computation.

**Summary Of The Review:**

I recommend accepting this paper for its technical novelty, the model acceleration, the performance gain (when it is trained with a long schedule), and the potential of improving more with a better attentive score.

---

> ### Author Response · Authors · 2021-11-21
> **Response to Reviewer WtKh (Part 1/2)**
>
> We would like to thank the reviewer for providing valuable comments and we answer the raised questions below.
>
> **1. Incorporating the methods into multi-scale transformers.**
>
> This is a good suggestion. Unfortunately, it requires substantial efforts to incorporate our method into multi-scale transformers. On the one hand, current multi-scale transformers [a,b] typically remove the [CLS] token as they focus on dense visual recognition tasks. Even though we can replace the [CLS]-to-tokens attentiveness criterion with the tokens-to-tokens attentiveness criterion as you have suggested (please see our response below), it is still nontrivial to incorporate EViT into multi-scale transformers because we need to keep track of the removed tokens in the spatial transformations (e.g., convolutions) in typical multi-scale ViTs [a,b]. A possible option is to use placeholders to replace the removed tokens in those spatial transformations and exclude the placeholders in fully connected layers. As the bulk of computation of multi-scale ViTs lies in fully connected layers [a,b], it can still save a considerable amount of computation by incorporating EViT in the multi-scale ViTs.
> On the other hand, the biggest usage of multi-scale ViTs is probably on dense prediction tasks such as object detection and semantic segmentation. Since EViT removes tokens in intermediate layers, it may render the spatial image representations incomplete in deeper layers and thus EViT may not be directly used in those tasks. It is an interesting and potential research direction as to how to incorporate dynamic token removal in ViTs in dense prediction tasks in future research.
>
>
> **2. Comparison of the vanilla inattentive token removal and inattentive token fusion.**
>
> We agree that the improvement upon using inattentive token fusion is small. However, the improvement is not affected by training noise. We report our ablation results in Table 2, where we train EViT for three times independently and average the evaluation results. We have also included the standard deviation of these three trials in Table 2. It shows that when compared to a vanilla model, there is a lower accuracy fluctuation by using inattentive token fusion.
> Our intuition of introducing inattentive token fusion is to recover the dropped tokens which are still able to contribute to the prediction results. Although the improvement is small, there is no additional computational overhead introduced. From another perspective, it indicates the effectiveness of our attentive token identification because the majority of effective tokens are well preserved.
>
>
> **3. Training DynamicViT from scratch.**
>
> We have trained DynamicViT from scratch for 300 epochs and included a comparison of the training and finetuning results of DynamicViT and those of EViT in Table 7 in the main manuscript. We have found that it is very difficult to train DynamicViT from scratch and we have encountered NaN issues in many initial trials even though we have followed the training configurations of DynamicViT. After trying many combinations of different learning rates, warmup epochs, and gradient clipping, we finally managed to avoid the NaN issue by extending the warmup to 150 epochs and using a gradient clipping with a threshold of 1. The results in Table 7 show the advantages of EViT over DynamicViT in both finetuning and training from scratch, and EViT outperforms DynamicViT in recognition accuracy under the same computational cost while EViT uses fewer parameters.
>
>
> **4. Using full tokens-to-tokens attention for attentive scores calculation.**
>
> This is an insightful comment. We actually conducted a few full tokens-to-tokens experiments in our initial codebase but it seemed less effective than the CLS-to-tokens strategy at that time. During the rebuttal, we have made further efforts to carefully investigate the effect of using full tokens-to-tokens attention for attentive scores calculation. Specifically, the full tokens-to-tokens attentive scores are computed by averaging the attention vectors of each token (i.e., averaging over all the rows of the attention map). We have reported the experimental results in Table 10 of Appendix B, which shows that EViT with the tokens-to-tokens attention strategy performs comparably with the CLS-to-tokens attention strategy while is less efficient (since the attentive scores are computed with the whole attention map). Please refer to the appendix for the implementation details of the full tokens-to-tokens strategy.
>
> **Reference**
>
> [a]. Ze Liu, Yutong Lin, Yue Cao, Han Hu, Yixuan Wei, Zheng Zhang, Stephen Lin, and Baining Guo. Swin transformer: Hierarchical vision transformer using shifted windows. In IEEE ICCV, 2021.
>
> [b]. Wenhai Wang, Enze Xie, Xiang Li, Deng-Ping Fan, Kaitao Song, Ding Liang, Tong Lu, Ping Luo, and Ling Shao. Pyramid vision transformer: A versatile backbone for dense prediction without convolutions. In IEEE ICCV, 2021.

---

> > ### Author Response · Authors · 2021-11-21
> > **Response to Reviewer WtKh (Part 2/2)**
> >
> > **5. Using DINO mask as an attentive region in EViT.**
> >
> > As suggested, we have performed an experiment to use the DINO mask as an attentive region in EViT and provided the result in Table 14. Specifically, we use the CLS attention produced by the last layer of DINO to guide the token selection process in EViT. Instead of starting the token reorganization process at the 4-th layer of EViT-DeiT-S as we have done in previous experiments, we start the token reorganization at the first layer. Specifically, we select the top 50% tokens that correspond to the positions with the strongest DINO attention (i.e., corresponding to the top 50% attention values in the DINO attention). Experimental results in Table 14 show that EViT works well with the DINO attention. As a comparison, when the reorganization layer is placed in the first layer in a vanilla EViT, as shown in Table 13, a significant accuracy reduction is observed, while EViT equipped with DINO can maintain its recognition ability even if 50% of tokens are removed as early as the first layer. However, this is not an efficient approach in practice because obtaining the DINO attention requires forwarding the input through the DINO model (i.e., a DeiT) and bringing extra computational burden, as shown by the "Combined MACs" in Table 14.
> >
> >
> > **6.  The interpretability of the learned mask in EViT.**
> >
> > In DINO, the authors evaluated the quality of the attention map by probing the self-attention map. Unfortunately, the code of probing the self-attention map is not available to the public, so we are unable to perform the same evaluation for the attention map as DINO did at present.
> >
> > On the other hand, from Figures 3 and 5, we can see that the masks produced by EViT are quite intuitive and of good quality, which means that the attention values of the CLS token to other tokens correlate well with the semantic segmentation of images. Therefore, we can deliberately use some low quality masks in EViT to see how its performance changes with the quality of the masks. Note that we use the largest CLS-to-tokens attention values to obtain the mask of the keeping tokens (we call this method EViT-topk). Thus, the worse possible mask one can obtain is probably by using the smallest CLS-to-tokens attention values (we call this method EViT-mink), and a not-so-bad mask can be obtained by randomly generating a mask without considering the CLS-to-tokens attention (we call this method EViT-random). We perform an experiment to see the classification accuracy of EViT under these circumstances. As shown in Table 15 in the appendix, the Top-1 accuracy of these three methods (keep rate=0.7) is: 79.5 (EViT-topk) > 78.4 (EViT-random) > 76.4 (EViT-mink). From the experimental results, we conclude that the performance of EViT indeed correlates with the quality of the masks and masks of better quality help improve the accuracy. Therefore, we believe it is a potential direction for future research as to how to (efficiently) obtain a better mask to improve the performance of EViT.
> >
> >
> > **7. The existence of inattentive tokens that are preferred in the deep layer of ViT.**
> >
> > We have visualized the attentive scores of the dropping tokens in a naively pre-trained DeiT-S in Figure 6 in the appendix. The attentive scores of some of the samples basically decrease monotonically while some of them first increase a bit and then decrease. Although there exist dropping tokens that are more preferred in the remaining layers, our experimental results refute the idea that those tokens are really important to the final prediction correctness. If they were important to tasks, then there would have been a significant accuracy reduction in, e.g., EViT-DeiT-S (with a keep rate of 0.7), because the token absence rate is as large as $1- 0.7^3= 0.657$ at the last layer. But the actual accuracy drop is just 0.3%. Thus, those later-more-preferred tokens are not essential to visual prediction at the remaining layers in EViT.

---

> > > ### Comment · Reviewer_WtKh · 2021-11-30
> > > **Thank you for the response**
> > >
> > > I thank the authors for the feedback and the modifications. My concerns 3 (DynamicViT) and 4 (tokens-to-tokens attention) are sufficiently addressed. Suggestion 1 is also taken into account and shows insightful results. Therefore, I would like to raise the score from 6 to 8.
> > >
> > > I also thank the author for the discussion on weakness 1 and I agree that this can be left to future work. The improvement of inattentive token fusion is still limited, as raised by reviewer [soMj] as well, but I don't think this concern can flip the decision.

---

### Official Review · Reviewer_y9ec · 2021-10-31

**Correctness:** 4
**Technical Novelty And Significance:** 4
**Empirical Novelty And Significance:** 4
**Recommendation:** 8
**Confidence:** 4

**Main Review:**

Pros:
The proposed method is simple while effective. Using the class token to guide image token selection seems reasonable and is well motivated by the observations (Fig. 1 and Table 1). The whole computation does not bring much time cost while reducing inattentive tokens
during the ViT feed forward process. This design benefits both offline and online stages.

In the experiments, the proposed method accelerates recent ViT models (DeiT and LV ViT) by increasing throughput and decreasing MAC. Meanwhile, the recognition accuracy does not drop much. Another perspective is that under a similar computational cost, using more tokens from higher resolution input images improves the prediction accuracies. This will benefit a series of ViTs without bringing additional time costs.

Cons:
The proposed method illustrates inattentive token removal at the 4th, 7th, and 10th layers. Whatis the rationale behind this choice? How to decide which layer to process when handling other ViT models? A clear illustration upon how to integrate this method into ViT models will be more convincing.

The token reorganization seems to fuse discarded tokens into one to supplement attentive tokens. As have explained in Sec. 3.3, this might be because the attentive token selection is not stable at the beginning of the training. If that is the case, a visualization upon token variations (initially random token selection, then the discarded token converges to the unrelated content) would be useful to support the claim.


**Summary Of The Paper:**

In this paper, an EVIT method is proposed for vision transformer speedup. It reduces image tokens based on the token attentiveness, which is measured by the class token. The inattentive tokens are reorganized as one to support attentive tokens. Experiments have shown on the
benchmarks for visual recognition.

**Summary Of The Review:**

Token processing is promising as it potentially benefits all the ViTs. The proposed method improves ViTs from the efficiency and accuracy perspectives. This will bring a wide range of impacts on the ViTs development. This reviewer recommends acceptance.

---

> ### Author Response · Authors · 2021-11-21
> **Response to Reviewer y9ec**
>
> We would like to thank the reviewer for providing valuable comments and we answer the raised questions below.
>
> **1. The rationale behind the choice of token reorganization locations.**
>
> To thoroughly investigate the influence of token reorganization locations, we have conducted extended experiments on the reorganization layers (locations and numbers) and the tokens keep rates. The results are shown in Table 13 in the appendix, which reveals two conclusions:
>
> 1. Moving the reorganization modules into shallower layers deteriorates the accuracy. For example, when the token reorganization module is placed before the third layer (i.e., at the first or second layer), the recognition accuracy drops considerably even though the computational cost is the same. This suggests that ViT cannot identify the important tokens at the early stages, which is quite reasonable as the processing of input tokens is just started at shallow layers, where the attention maps are unreliable for token removal.
> 2. Placing the reorganization modules in different deeper layers has only marginal influence on the accuracy. For instance, when the token reorganization modules are placed behind the third layer, the resulting models have basically the same accuracy. This suggests that EViT has stable performance as long as the reorganization locations are not in shallow layers.
>
> Because of these reasons, we refrain from searching for a (slightly) better configuration of the reorganization locations and keep rates. Instead, we adopt a simple strategy to decide the reorganization locations in our experiments, where the token reorganization locations cut the ViT into blocks with the same number of layers. Specifically, for a ViT with $L$ layers and $t$ token reorganization layers in total, we first obtain the separating length $s = L / (t+1)$. Then, the layer indices of the reorganization layers are $[s+1, 2s+1, \dots, ts + 1]$, which cuts the ViT evenly. For the keep rates, we simply set them to the same value for each token reorganization module in the EViT.
>
>
> **2. Visualization of the evolution of the inattentive tokens.**
>
> We have added a visualization of the evolution of inattentive tokens during the EViT training process. Please see Figure 7 in the appendix in our revised manuscript. It can be observed that at the initial training stage (e.g., the first epoch), the EViT model tends to identify the inattentive tokens randomly. As the training proceeds, the discarded tokens gradually converge to the less informative contents (e.g., the backgrounds).

---

### Official Review · Reviewer_SSye · 2021-11-02

**Correctness:** 4
**Technical Novelty And Significance:** 4
**Empirical Novelty And Significance:** 4
**Recommendation:** 8
**Confidence:** 4

**Main Review:**

PROS:
+	Image token identification via the class token guidance is interesting. The class token adaptively removes image tokens based on the corresponding category influence. The discarded tokens are reorganized/fused into one token to facilitate network training at the initial training stages.
+	The experiments on the efficiency and accuracy evaluations show the proposed method is effective for current Vit models.
CONS:
-	The proposed method gradually removed tokens from different layers of ViT models. Is there any relationship between these removed tokens from different layers? Since the removed tokens are hardly maintained, will the proposed fusion strategy be able to restore them at the remaining layers?
-	Given a ViT model, is the design of the proposed method arbitrary for all the layers or some fixed layers? Where to drop and fuse tokens within a ViT model deserves further discussion.
-	What leads to the difference between Table 1 and Table 2(a)? A straightforward inattentive token fusion removal decreases the DeiT more, while without inattentive token fusion decrease the same model less? If the rate of Table 2(a) is set as 0, will the results be the same?
-	Reporting the actual time cost will benefit a thorough understanding of ViTs acceleration.

**Summary Of The Paper:**

Research on Vision Transformer s ( ViTs is heavy Different from the prior investigations that focus on proposing ViTs structures, this work focuses on image tokens and study how to effectively wipe out useless ones to model acceleration This brings a potential advantage that
when maintaining the same time cost, models are able to take more tokens i.e, from a higher resolution image ) for prediction accuracy improvement. These two aspects are evaluated in the experiments.

**Summary Of The Review:**

The token reorganization is new. The results are fine. Addressing these raised issues above will make this submission more convincing.

---

> ### Author Response · Authors · 2021-11-21
> **Response to Reviewer SSye (Part 1/2)**
>
> We would like to thank the reviewer for providing valuable comments and we answer the raised questions below.
>
> **1. The proposed method gradually removed tokens from different layers of ViT models. Is there any relationship between these removed tokens from different layers? Since the removed tokens are hardly maintained, will the proposed fusion strategy be able to restore them at the remaining layers?**
>
> The set of the removed tokens from early layers is a subset of that in the subsequent layers. Besides, the removed tokens in later layers are also semantically similar and spatially close to those removed in early layers. This is especially evident by looking at the evolution of the removed tokens during the EViT training process. As shown in Figure 7 in the appendix, the removed tokens tend to spread out at early training stages, and then they tend to converge to spatially close regions in the image as training proceeds. The spatially close regions typically share similar semantic information (e.g., sky, grass, etc.).
>
> In our proposed method, once a token is removed/fused at an intermediate layer, it will not ever be used in the subsequent layers. Thus, the proposed fusion strategy is unable to restore the removed tokens in the remaining layers. However, a recently released paper [a] on masked autoencoders reveals an interesting aspect of the reconstruction capacity of ViTs, where the authors show that ViTs are able to reconstruct as much as 75% masked input tokens in the output layers based on the information provided by the 25% tokens left. Although the reconstructed images are a bit blurry, it still shows the possibility of (partially) restoring the removed tokens in ViTs. We consider this an interesting direction in future research.
>
>
> **2. Choosing the layer for token reorganization.**
>
> To thoroughly investigate the influence of token reorganization locations, we have conducted extended experiments on the reorganization layers (locations and numbers) and the tokens keep rates. The results are shown in Table 13 in the appendix, which reveals two conclusions:
>
> 1. Moving the reorganization modules into shallower layers deteriorates the accuracy. For example, when the token reorganization module is placed before the third layer (i.e., at the first or second layer), the recognition accuracy drops considerably even though the computational cost is the same. This suggests that ViT cannot identify the important tokens at the early stages, which is quite reasonable as the processing of input tokens is just started at shallow layers, where the attention maps are unreliable for token removal.
> 2. Placing the reorganization modules in different deeper layers has only marginal influence on the accuracy. For instance, when the token reorganization modules are placed behind the third layer, the resulting models have basically the same accuracy. This suggests that EViT has stable performance as long as the reorganization locations are not in shallow layers.
>
> Because of these reasons, we refrain from searching for a (slightly) better configuration of the reorganization locations and keep rates. Instead, we adopt a simple strategy to decide the reorganization locations in our experiments, where the token reorganization locations cut the ViT into blocks with the same number of layers. Specifically, for a ViT with $L$ layers and $t$ token reorganization layers in total, we first obtain the separating length $s = L / (t+1)$. Then, the layer indices of the reorganization layers are $[s+1, 2s+1, \dots, ts + 1]$, which cuts the ViT evenly. For the keep rates, we simply set them to the same value for each token reorganization module in the EViT.
>
>
> **3. The differences between Table 1 and Table 2(a).**
>
> We are sorry, but there might be a slight misunderstanding with Table 1. We simply use the pre-trained model provided by DeiT [b] for inference to obtain the results in Table 1. Note that the parameters of an EViT are exactly the same as the corresponding ViT model. Therefore, we can easily load the parameters of a pre-trained DeiT into EViT and then run the EViT in ImageNet testing. So no training of EViT is involved in Table 1, and there is only testing. In Table 2(a), the EViT model is trained from scratch with the proposed token reorganization strategy. Therefore, the EViT in Table 2(a) can adaptively learn in the absence of some inattentive tokens while the EViT in Table 1 does not get to learn to adapt to the token removal scenario. The comparison between Table 1 and Table 2(a) shows that incorporating training into EViT is important and validates the effectiveness of the proposed methods. If the keep rate in Table 1 and Table 2(a) is set to 1, the results should be the same.
>
> **Reference**
>
> [a]. Kaiming He, Xinlei Chen, Saining Xie, Yanghao Li, Piotr Dollar, and Ross Girshick. Masked Autoencoders Are Scalable Vision Learners. arXiv preprint arXiv:2111.06377, 2021.
>
> [b]. https://github.com/facebookresearch/deit

---

> > ### Author Response · Authors · 2021-11-21
> > **Response to Reviewer SSye (Part 2/2)**
> >
> > **4. The results of actual time cost.**
> >
> > Thanks for the suggestions. We provide a training time comparison of EViT and vanilla ViT in Table 11 in the revised manuscript, which shows that EViT requires a considerably smaller GPU\*hours to train when compared to a vanilla DeiT under the same number of epochs.
> > For the actual time cost for ViT inference, we can calculate the time cost from the throughput of the model. For example, for DeiT-S with an inference throughput of 2923 images/s, the actual time cost to obtain the prediction outcome for a single image is 1/2923=0.00034s =0.34 ms.

---

> > > ### Comment · Reviewer_SSye · 2021-12-02
> > > **Rating**
> > >
> > > I would like to thank the authors for their response. My minor concerns have been addressed, so I keep my rating.

---

### Comment · Area_Chair_SGog · 2021-11-09
**comparison to a related paper?**

It will be great if the authors could also conceptually compare their approach against a recent similar work: https://arxiv.org/abs/2106.11297

I observe that the detailed approaches are different, but it will be meaningful to explicitly compare as they seem to share similar motivation and design?

---

> ### Author Response · Authors · 2021-11-21
> **Response to Area Chair SGog**
>
> We would like to thank the AC for pointing out this related article (i.e., TokenLearner [a]). We have provided a conceptual comparison between their approach and ours in Section 2.2 in the revised manuscript. The relationship is summarized as follows:
>
> ### Similarity
> - Both TokenLearner and our EViT accelerate Vision Transformers by reducing the number of image tokens. Meanwhile, the two methods improve image recognition accuracy based on each proposed method separately.
>
> ### Differences
> 1. The core designs of token reduction between TokenLearner and EViT are different. TokenLearner introduces convolutional layers to produce spatial attention weight maps. A weight map combines image pixels into a single token. These convolutional layers are additional modules for ViTs and the learnable convolutional parameters inevitably bring inductive bias. In contrast, our EViT identifies token attentiveness via lightweight attention computation between the class tokens and the Key matrices in attention heads. We do not introduce additional parameters or modules and our attention computation is naturally integrated into ViTs during training and inference.
> 2. The rationales behind image classification improvement are different. TokenLearner essentially introduces a different tokenization method for image classification improvement. Their rationale is to increase network capacity while maintaining computational speed. In contrast, our EViT does not alter the ViT network but increases the number of input image tokens with higher resolution images. There is no additional network design in our EViT for performance improvement.
> 3. The application scenarios are different. TokenLearner proposes a general method for both image and video ViT acceleration and focuses on video understanding. In contrast, our EViT is specifically designed for image ViTs. We use class tokens to identify related image tokens and fuse the inattentive tokens for end-to-end training. Nevertheless, extending EViT to video transformers is interesting and we will conduct this research in our future work.
> 4. The training configuration is different. The JFT-300 dataset is utilized for TokenLearner pretraining and ImageNet-1K for finetuning. The pretraining on JFT-300 (300M images) brings huge computation requirements. In our EViT, we only use ImageNet-1K to train from scratch.
>
> Reference
>
> [a]. Michael S Ryoo, AJ Piergiovanni, Anurag Arnab, Mostafa Dehghani, and Anelia Angelova. Tokenlearner:  Adaptive space-time tokenization for videos. In NeurIPS, 2021.

---

### Author Response · Authors · 2021-11-21
**General Response and Summary of Revisions**

We would like to thank the Area Chair and all the reviewers for providing insightful comments and constructive feedback to our ICLR submission. To address the concerns raised by the reviewers, we have substantially extended the experiments and included a thorough discussion of the results in the revised manuscript. We have also included a summary of changes to the revised manuscript here. We have highlighted the revisions in our main manuscript in blue (but not those in the appendix, as most of them are new). Section B in the appendix is newly added in our revised paper.


### Summary of Revisions

- To address the concerns raised by Reviewers [WtKh] and [soMj], we have made further efforts to compare the two variants of EViT, namely, vanilla inattentive token removal and inattentive token fusion, to demonstrate the advantages of the latter one. Specifically, using the strategy of inattentive token fusion provides slightly higher accuracy with lower fluctuation across repeated experiments, as shown in Tables 2 and 9.

- We have tried training DynamicViT from scratch and provided a comparison between DynamicViT and the proposed EViT in Table 7 in the main manuscript, as suggested by Reviewer [WtKh]. The results show the advantages of our approach over DynamicViT.

- As suggested by Reviewer [WtKh], we have conducted an ablation study on using the tokens-to-tokens attentive scores as a token removal criterion in Table 10 in the appendix.

- As suggested by Reviewer [SSye], we have included a report of the training time of EViTs and vanilla ViTs in Table 11 in the appendix, which shows that EViTs require considerably less time to train than vanilla ViTs.

- We have conducted more experiments on finetuning EViT on high resolution images in Table 12 and discussed the results in the appendix.

- We have moved the study of token reorganization locations into Table 13 in the appendix as it has been substantially extended, where more configurations of token reorganization locations and keep rates have been examined and more discussions have been added, as suggested by Reviewers [SSye, y9ec, and soMj].

- We have tried using the DINO attention map as a criterion for inattentive token removal, as suggested by Reviewer [WtKh], and reported the results in Table 14 in the appendix.

- We have performed a study of the correlation between the quality of the attention masks and the performance of EViT in the appendix, where a random token keeping strategy and a min-k token keeping strategy have been reported and compared to the proposed top-k keeping strategy in Table 15.

- We have provided a visualization of the evolution of the attentive tokens during the training process of EViT in Figure 7 in the appendix, which shows the tokens that are kept in EViT converge to semantically meaningful patches in the images, as suggested by Reviewer [y9ec].

- We have added a description of the differences between our method and the methods depicted in Figure 4 in the main manuscript, and have compared EViT to another downsampling operation, i.e., PVT, as suggested by Reviewer [soMj].

- We have included a discussion of the differences between TokenLearner and EViT in Section 2.2 in the revised manuscript, as suggested by the Area Chair.

- We have carefully revised the manuscript and corrected all the typos, as suggested by Reviewer [soMj].

We sincerely hope the revisions and responses have made the contributions of our work clearer and addressed the concerns raised by the reviewers.

---

### Public Comment · ~Muzammal_Naseer1 · 2022-02-20
**Please Consider Citing our Work Relevant to Masking on Objects and Backgrounds**

Hi,

Congratulations on the spotlight acceptance. The results are really interesting.

I would like to bring your attention to our recent Neurips work [1] (https://arxiv.org/abs/2105.10497) that also thoroughly discusses ViTs robustness against salient and non-salient masking along with other intriguing properties.

It would be great if authors can cite our work [1] considering both EViT and [1] observe the same phenomenon and draw similar conclusions regarding ViT's robustness against salient /non-salient masking.

[1] Intriguing Properties of Vision Transformers

---

### Decision · Program_Chairs · 2022-01-20

**Decision:**

Accept (Spotlight)

**Comment:**

The paper presents an approach to select visual tokens in images and reorganize them for the object classification, within Transformers. All four reviewers find the paper interesting and novel, and they are also very positive about the experimental results. The authors also addressed minor concerns of the reviewers successfully through the discussion phase, clarifying details and adding experiments.

We recommend accepting the paper.